# Dynamic Inhomogeneous Quantum Resource Scheduling with Reinforcement Learning

## Abstract

A central challenge in quantum information science and technology is achieving real-time estimation and feedforward control of quantum systems. This challenge is compounded by the inherent inhomogeneity of quantum resources, such as qubit properties and controls, and their intrinsically probabilistic nature. This leads to stochastic challenges in error detection and probabilistic outcomes in processes such as heralded remote entanglement. Given these complexities, optimizing the construction of quantum resource states is an NP-hard problem. In this paper, we address the quantum resource scheduling issue by formulating the problem and simulating it within a digitized environment, allowing the exploration and development of agent-based optimization strategies. We employ reinforcement learning agents within this probabilistic setting and introduce a new framework utilizing a Transformer model that emphasizes self-attention mechanisms for pairs of qubits. This approach facilitates dynamic scheduling by providing real-time, next-step guidance. Our method significantly improves the performance of quantum systems, achieving more than a $3\times$ improvement over rule-based agents, and establishes an innovative framework that improves the joint design of physical and control systems for quantum applications in communication, networking, and computing.

## 1 Introduction

Quantum Information Science (QIS) is an emerging field poised to revolutionize computation, communication, precision measurement, and fundamental quantum science. At the heart of QIS lies the quantum resource state, which underpins quantum information representation and processing. For this paper, a quantum resource state refers to an entangled network of qubits (Appendix A.5, A.6). Achieving larger, high-fidelity quantum resource states is critical for advancing applications in material and drug discovery, optimization, and machine learning via quantum computing (Appendix A.6). Scaling physical qubit resources to meet the demands of quantum information processing is increasingly enabled by advances in solid-state quantum systems such as color centers and quantum dots (Appendix A.3). These systems leverage modern semiconductor fabrication technologies and heterogeneous integration (Wan et al., 2020; Li et al., 2024; Clark et al., 2024; Golter et al., 2023; Starling et al., 2023; Palm et al., 2023). Such technologies allow for large-scale quantum systems with dynamically configurable qubit interactions through remote entanglement (Humphreys et al., 2018), customized to meet system requirements (Choi et al., 2019; Nickerson et al., 2014; Nemoto et al., 2014). However, optimizing the control and scheduling of these large, complex systems is essential to maximize performance. Quantum resources exhibit inherent inhomogeneity due to their distinct physical properties and control mechanisms, which vary spatially and temporally. This inhomogeneity, coupled with the probabilistic nature of quantum operations like heralded remote entanglement (Appendix A.10), introduces stochastic challenges in error detection and system performance. These complexities render the optimization of quantum resource state construction an NP-hard problem. Nevertheless, achieving larger, high-fidelity quantum resource states offers exponential advantages in quantum information processing.

Recent developments in reinforcement learning have demonstrated significant value in various scientific and technological fields. This includes advances in protein structure design (Wang et al., 2023b), mathematics discovery (Fawzi et al., 2022), chip design (Mirhoseini et al., 2021), optimized control within the laboratory (Degrave et al., 2022; Szymanski et al., 2023). In addition, the

application of machine learning in quantum technologies is becoming increasingly critical (Metz & Bukov, 2023; Chen et al., 2022; Mills et al., 2020; Sels et al., 2020; Carrasquilla et al., 2019; Lu & Ran, 2023). However, leveraging these advances in a scalable quantum engineering system requires a system-level optimization approach. This approach needs to take into account the varied properties of qubit arrays to effectively manage control and scheduling tasks.

In this paper, we formulate the quantum resource scheduling problem in a digitized environment with Monte Carlo Simulation (MCS) (Appendix A.9, B). This environment enables us to develop a rule-based greedy heuristic method that significantly outperforms the random scheduling baseline. We also train reinforcement learning agents in this interactive probabilistic environment. We introduce a "Transformer-on-QuPairs" framework that uses self-attention on inhomogeneous qubit pairs' sequential information to provide the dynamic, next-step scheduling guidance for qubit resource state building. Furthermore, this Transformer-on-QuPairs scheduler enhances quantum system performance by more than $3\times$ compared to our rule-based method in our inhomogeneous simulation experiment.

The remainder of the paper is organized as follows: Section 2 discusses related works. Section 3 defines our problem of dynamic inhomogeneous quantum resource scheduling, analyzes its complexity, and provides a benchmarking and scheduling example. In Section 4, we detail the RL-based optimization framework and Transformer-on-QuPairs architecture for dynamic scheduling strategies. Section 5 outlines the experimental setup and presents a comparison of the results. Section 6 concludes the paper, and Section 7 discusses the broader impacts of this work.

## 2 RELATED WORKS

Machine learning has helped the development of quantum information processing. The Transformer model, for example, has been effectively used in various applications such as quantum error correction (Wang et al., 2023a), quantum state representation using tensor networks (Chen et al., 2023; Zhang & Di Ventra, 2023), and quantum state reconstruction (Ma et al., 2023), and quantum error-correction code decoding (Bausch et al., 2024). Reinforcement learning has similarly found extensive application in a broad spectrum of quantum computing tasks, including quantum circuit design search (Herbert & Sengupta, 2018; Alam et al., 2023; Fösel et al., 2021; Pirhooshyaran & Terlaky, 2021), quantum architecture search (Kuo et al., 2021; Ostaszewski et al., 2021), quantum ground state identification (Mills et al., 2020), quantum control optimization (Lu & Ran, 2023; Metz & Bukov, 2023).

Previous studies have primarily focused on quantum computing platforms with limited qubit connectivity, such as those using superconducting circuits (Arute et al., 2019). In contrast, platforms that support all-to-all connectivity can utilize different protocols, such as cluster state quantum computing, offer greater control flexibility, allowing for enhanced optimization through machine learning. This paper explores a quantum control architecture tailored to a unique class of quantum resources featuring a spin-photon (Appendix A.8, Fig.6) interface conducive to remote entanglement routing. This setup affords a high degree of freedom in dynamic quantum resource scheduling, a domain where control strategies remain underexplored. The qubit platform based on the spin-photon interface has the potential for rapidly scaling, such as the heterogeneous integration between the diamond color center with the CMOS backplane (Li et al., 2023), PIC backplane (Wan et al., 2020), the T center in Si (Higginbottom et al., 2022), and the quantum dot (Coste et al., 2023) platform. These capabilities position it well for applications in quantum networking, communication, and computing. This control protocol is also compatible with the leading quantum platform with massive programmable connectivity such as the trapped ion (Srinivas et al., 2021), neutral atom array (Periwal et al., 2021), manufacturable photonic qubit (Alexander et al., 2024), and the hybrid systems encompassing diverse physical qubits (Mirhosseini et al., 2020).

## 3 DYNAMIC INHOMOGENEOUS QUANTUM RESOURCE SCHEDULING

In this section, we formulate the dynamic inhomogeneous quantum resource scheduling problem in graph representation. We begin by presenting an analysis of the problem's complexity and define benchmarks for assessing quantum system performance. Additionally, we develop a simulated

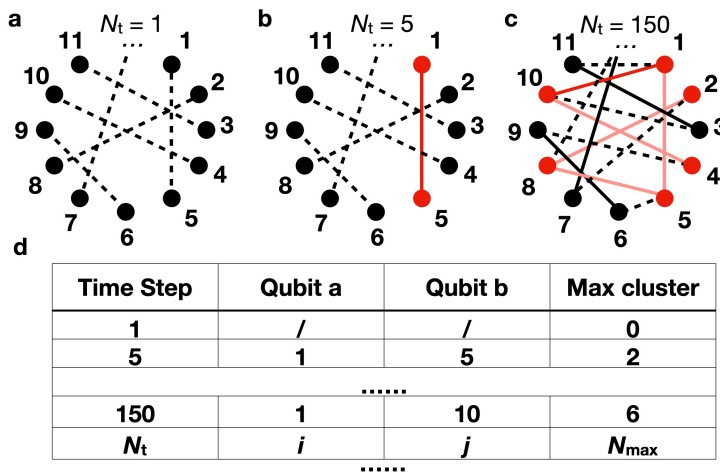

Figure 1: **Dynamic quantum resource scheduling game.** **a**, At the initial time step $N_t = 1$, individual qubit resources (represented by black circles) are depicted, poised for the formation of entanglement pairs (illustrated by dashed black lines). This figure shows only a portion of the 11 qubit nodes of the quantum resource. **b**, By the early stage at $N_t = 5$, each time step carries a probability of successfully establishing entanglement pairs. Newly formed entanglements are indicated by solid red lines, and the largest connected subgraph is highlighted with red nodes. **c**, At a later stage, $N_t = 150$, the diagram shows a larger connected qubit cluster within the quantum resource, with earlier entanglements depicted in lighter red. **d**, The result table records each successful entanglement event derived from the quantum simulation. For each time step $N_t$, when the entanglement between Qubit i and Qubit j is successfully established through Monte Carlo simulation, a new entry is added to the table, updating the maximum size of the connected graph $N_{\max}$.

experiment example of dynamic quantum resource scheduling and describe the methods used to generate assumed system pre-information.

**Complexity of the cluster building scheduling problem** Before delving into the specifics of our quantum cluster building scheduling challenge, it is useful to examine a related, yet simpler, NP-hard problem known as the Minimum Weight Connected Subgraph Problem (MWCSP) (Haouari et al., 2013; Hwang et al., 1992). This problem is defined on a weighted graph $G = (V, E)$, where each edge is assigned a real number weight through the function $\omega$. The goal is to find a connected subgraph $H = (V', E')$, with $V' \subseteq V$ and $E' \subseteq E$, that minimizes the total weight of the edges $\omega(E')$, ensuring connectivity among all vertices in $V'$. In our quantum context, this scheduling challenge equates qubits and their entanglement links to the vertices and edges of the graph, respectively, with $\varepsilon_{ij}$ representing the quantum error associated with each link. Our objective is to minimize the total quantum error $\varepsilon = \sum_{ij} \varepsilon_{ij}$ (total weight of the edges $\omega(E')$) across a connected cluster of $|V'|$ qubits ($V' \subseteq V$, where $V$ is the whole qubit vertex set). The complexity intensifies in what we term the dynamic MWCSP, which incorporates entanglement links that not only have a success rate $p_{\text{succ}} \leq 1$ but also allow the establishment of $K_{\text{pairs}} \geq 1$ entanglements simultaneously. When $p_{\text{succ}} = 1$ and $K_{\text{pairs}} = 1$, this dynamic variant reduces to the standard MWCSP, underscoring that our dynamic MWCSP is at least as challenging as the NP-hard baseline. Constructing a cluster state that includes an $N$-qubit cluster within this framework causes the search space for the dynamic MWCSP to expand exponentially, scaling beyond $O(N^N)$ (Cayley, 1878).

**Quantum system performance benchmarking** The effectiveness of a cluster state quantum system is measured by the cluster-state quantum volume (Cross et al., 2019) $V_Q$, which is defined as $\mu = \log_2 V_Q = \operatorname{argmax}_{n \leq N} \min\left(n, \frac{1}{n\varepsilon}\right)$. This metric applies to our cluster system when interfaced with general quantum computing hardware. In this context, $n$ refers to the number of qubits in the cluster, $\varepsilon$ is the total error throughout the quantum cluster, and $N$ indicates the maximum cluster resource available. Given that each interconnection error within our qubit cluster nodes

$V$ is considerably low ($\varepsilon_{ij} \ll 1$), the aggregate cluster error can be calculated as $\varepsilon = \sum_{ij} \varepsilon_{ij}$. This equation influences the determination of the maximum sustainable cluster size $N_{\max}$ and the cumulative cluster error $\varepsilon$, both parameters that typically grow with increasing scheduling time steps $N_t$ in the system.

**Dynamic quantum resource scheduling example**    We used Monte Carlo simulations to depict the cluster building process, as demonstrated in Fig.1. The simulation environment has $N_q$ qubit resources and has maximum $N_q/2$ entanglement workers to attempt entanglement in parallel. Each time step for attempting entanglement has a success probability $R_{ij} \leq 1$ between the qubits $i$ and $j$. When an entanglement is successfully established, its details are recorded in a progress table, which keeps track of the success time step index $N_t(i, j)$ and updates the qubit graph using a disjoint-set data structure. The maximum cluster size achieved, $N_{\max}$, is also recorded. In a scenario targeting a 40-qubit system (illustrated in Fig.3), updates to the progress table cease once $N_{\max}$ exceeds 30, in alignment with our predefined error metrics. Data for Figs. 3b and 3c are subsequently extracted from these progress table entries. The error associated with each established entanglement is calculated using the formula $1 - F_{ij} \exp(-\Delta t_{ij}/T_{\mathrm{mem}})$, where $F_{ij}$ is the fidelity and $\Delta t_{ij}$ is the time elapsed since the formation of the entanglement, calculated as $(N_t - N_t(i, j))/r_{\mathrm{ent}}$ within the $N_t$ trial time steps. $T_{\mathrm{mem}}$ is the coherence time of the qubit memory, and $r_{\mathrm{ent}}$ is the rate of entanglement attempt.

**Quantum system pre-information generation**    To support the simulations of the cluster state building process, we generate random performance distributions for $F_{ij}$ and $R_{ij}$. Here, $F_{ij}$, which denotes the fidelity of the entanglement, is determined using a Gaussian distribution with a mean $\bar{F} = 0.98$ and a standard deviation $\sigma(F)$. Fidelity values $F_{ij}$ that surpass the maximum allowable fidelity, $\max(F) = 0.998$ (IBM, 2024), are adjusted to this upper limit. Likewise, the success probability $R_{ij}$ for forming each entanglement is derived from a Gaussian distribution centered on $\bar{r}$ with a standard deviation $\sigma(r)$. Both $F_{ij}$ and $R_{ij}$ parameters can be produced through Quantum Monte Carlo Simulation (QMCS), utilizing characterized experimental data as detailed in the Appendix B. This approach ensures that the scheduling strategy is not only theoretically sound but also practically feasible for implementation on actual quantum information processing systems.

## 4    A REINFORCEMENT LEARNING FRAMEWORK

In this section, we present our reinforcement learning (RL)-based optimization framework along with a detailed explanation of the Transformer-on-QuPairs architecture.

**RL-based optimization framework**    Figure 2a presents an RL-based framework designed to enhance the overall performance of a quantum system. The process starts with pre-characterized system information, which includes matrices for entanglement fidelity ($\boldsymbol{M}_{\mathrm{F}}(i, j) = F_{ij}$) and success rate ($\boldsymbol{M}_{\mathrm{R}}(i, j) = R_{ij}$).

The RL agent receives the state matrix ($\boldsymbol{M}_{\mathrm{S}}$) as input and generates an output action matrix ($\boldsymbol{M}_{\mathrm{A}}$). Each element in this matrix represents the potential cost associated with selecting that particular action within the strategy scheduler. The strategy scheduler selects the action with the lowest cost from the action matrix for the available operation qubits and forwards this to the heralding entanglement worker, subsequently updating the state matrix ($\boldsymbol{M}_{\mathrm{S}}$). $\boldsymbol{M}_{\mathrm{S}}$ is with size $N_q$ by $N_q$ as shown in Figure 2a. It uses the adjacent matrix to store whether two qubits are already entangled. If entangled, $\boldsymbol{M}_{\mathrm{S}}(i, j) = \boldsymbol{M}_{\mathrm{S}}(j, i) = 1$, else 0. A state check function ($f_1$) determines if the scheduling event is complete; if not, scheduling continues with the updated state matrix. Otherwise, the process transitions to a Monte Carlo simulation for each time step in the entanglement trial.

Successful entanglements alter the state matrix, and these modifications are tracked to evaluate against stopping conditions. If the cluster size exceeds a specified threshold, the system proceeds to calculate the reward, using the data to compute the $V_Q$ of the cluster state. If the cluster size remains below the threshold, another function ($f_2$) checks if enough idle qubits are available for subsequent scheduling. If conditions are met, the scheduling loop recommences, facilitated by the RL agent.

The RL agent, capable of dynamically adjusting to any number of qubits, employs a Transformer architecture (depicted in Fig. 2b). This architecture calculates a cost estimate matrix for potential links, utilizing preliminary information ($\boldsymbol{M}_{\mathrm{F}}, \boldsymbol{M}_{\mathrm{R}}, \boldsymbol{M}_{\mathrm{S}}$).

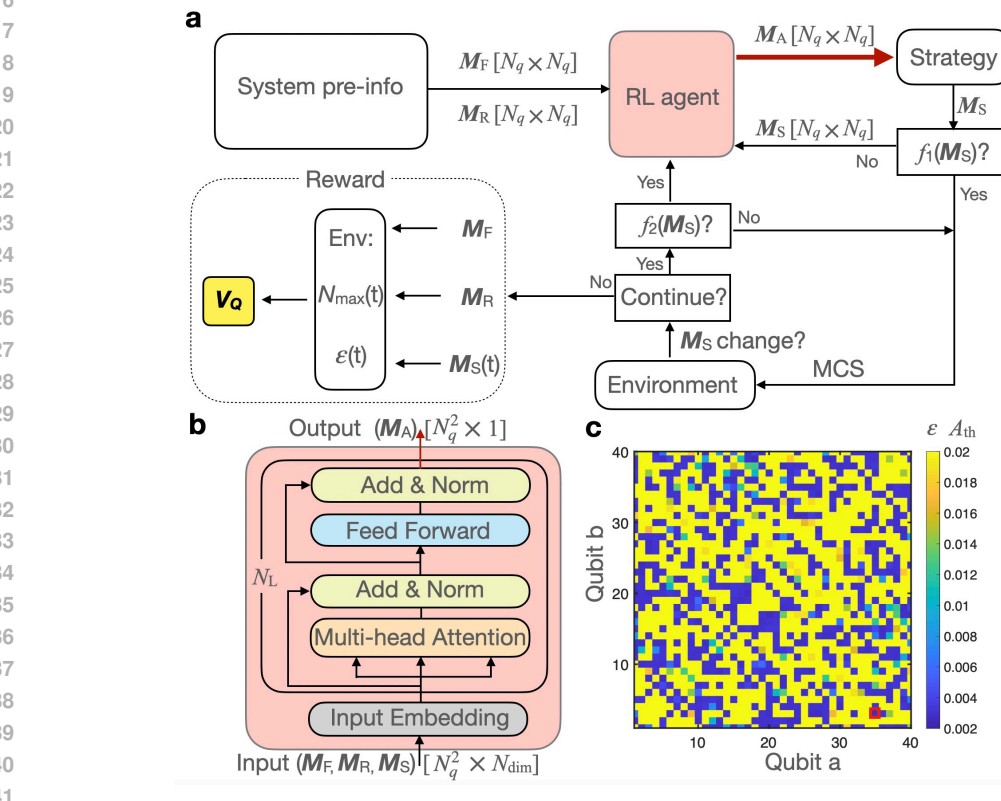

Figure 2: **RL-based optimization framework and dynamic scheduling strategies using the Transformer-on-QuPairs architecture. a,** The entire optimization flow aimed at enhancing $V_Q$ within a quantum system, starting with inputs from system pre-information data. **b,** Representation of the Transformer architecture used as an RL agent in **a**, processing a sequence of qubit pairs with input length $N_q^2$ and feature dimensions $N_{\text{dim}}$. It outputs a sequence predicting the cost function for each qubit pair, formatted as $N_q^2 \times 1$. **c,** The output of the transformer is further processed into a matrix to determine the minimal error ($\varepsilon$) for the operations in the next step. This processed action matrix sets an error threshold at $A_{\text{th}} = 0.02$. The suggested scheduling action, marked by a red rectangle, indicates the qubit pair with the minimum predicted error.

**Transformer-on-QuPairs architecture definition**    In this study, we utilize the standard Transformer architecture to model entanglement link creation within a quantum system. This architecture processes input tokens representing potential entanglement links. We encapsulate all possible entanglement combinations in an adjacency matrix for $N_q$ qubits, setting the input sequence length to $N_q^2$.

Each input token is a vector with dimension $N_{\text{dim}} = 7$, normalized between 0 and 1, composed of pre-information encoding, dynamic encoding, and position encoding:

- **Pre-information Encoding:** Utilizes three dimensions to express the fidelity ($F$), the exponential decay $\exp(-1/Rr_{\text{ent}}T_{\text{mem}})$, and the corresponding error term $1 - F\exp(-1/Rr_{\text{ent}}T_{\text{mem}})$.

- **Dynamic Encoding:** Reflects the current status of each entanglement link, derived from the adjacency matrix, indicating whether a link is established or pending.

- **Position Encoding:** Assigns normalized indices to the qubits involved in entanglements, represented as $i/N_q$ and $j/N_q$.

These vectors are embedded into a 32-dimensional space through an embedding layer. Following a bi-directional architecture, the Transformer predicts the real quantum error for each entanglement link, aiding the scheduling algorithm in selecting the link with the lowest anticipated quantum error.

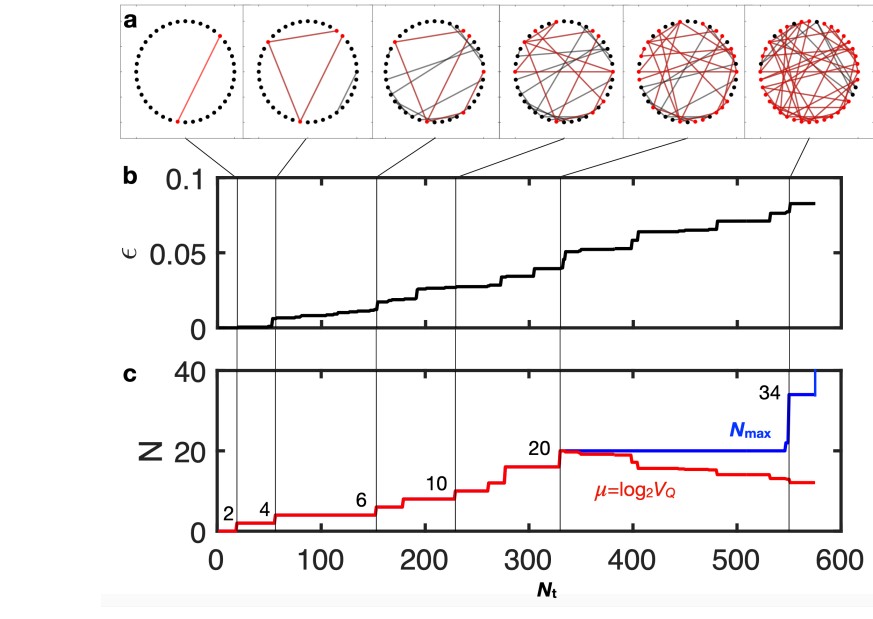

Figure 3: **Schematic of cluster state construction example** ($N_q = 40$). **a,** The scheduling simulation progress example at various time steps ($N_t$). Black circles represent individual qubit resources, red-labeled circles and connected lines indicate the largest subgraph formed among the qubits, and grey lines show the established entanglements between qubit nodes. **b,** Accumulation of errors ($\epsilon$) during the cluster state construction, plotted against the simulation time steps ($N_t$). The black connected line corresponds to the $N_t$ time step shown in panel **a**. **c,** Representation of the maximum number of connected subgraph size $N_{\max}$ as it evolves with $N_t$ (blue line), alongside the logarithm of the system's quantum volume ($\mu = \log_2 V_Q$), which also progresses with $N_t$ (red line)

The decision for subsequent actions is guided by the action matrix $M_A$, which combines the error matrix with $1 - F \exp(-1/R r_{\text{ent}} T_{\text{mem}})$ and the neural network's output weighted at 0.1. We set a threshold of $A_{\text{th}} = 0.02$ for $M_A$ to prioritize the better half of the high-quality entanglement links. The post-processed $M_A$ is depicted in Fig. 2c, with the red rectangle highlighting the next prioritized entanglement link for scheduling. If the only available options are above the threshold error $A_{\text{th}}$, the system opts to temporarily idle that corresponding entanglement worker.

## 5 EXPERIMENTS

We detail the experimental setups for rule-based and RL-based strategies below, along with the presentation of the experimental results. The relevant code and data are accessible with the instructions provided in the supplementary materials submitted.

### 5.1 EXPERIMENTAL SETUP AND METHODOLOGICAL COMPARISON

**Ruled-based strategies** We formulate the quantum resource scheduling problem in a digitizied environment with Monte Carlo simulation allowing for the implementation of rule-based scheduling strategies. Our baseline involves random scheduling, where we disregard the heterogeneous properties of qubit pairs and select the next scheduling step randomly using a generated action matrix. We compare this with the static minimum spanning tree (MST) approach, which is anticipated to be an effective heuristic when the quantum system's coherence time is indefinitely long or has a deterministic success probability during entanglement attempts. For quantum systems with limited coherence times that necessitate rapid dynamic actions, we employ a greedy algorithm that consistently selects the qubit pair with the lowest quantum error for the next step in resource state building scheduling.

Table 1: **Comparison of scheduling strategies: Rule-based vs. RL-based.** The superior system performance of the Transformer-on-QuPairs strategy compared to various rule-based approaches.

| Types | Strategies | $\bar{\mu}$ |
|---|---|---|
| Rule-based | Random | 3.85±0.23 |
| | Static Minimum Spanning Tree | 10.51±0.55 |
| | Greedy-on-QuPairs | 13.90±0.62 |
| RL-based | Transformer-on-Qubit | 3.91±0.31 |
| | Fully-connected-on-QuPairs | 14.70±0.72 |
| | Transformer-on-QuPairs | **15.58±0.84** |

**RL-based strategies** In addition to rule-based strategies, we employ RL-based strategies (Paszke et al., 2019) within the digitized environment, utilizing the capabilities of the Transformer-on-QuPairs agent and a fully-connected (FC) neural network. The Transformer-on-Qubit, with 1 layer, an embedding dimension of 320, a single attention head, and a feed-forward network hidden dimension of 640, takes qubit information as input to predict the most suitable qubit pairs for the next scheduling step. This process involves running the Transformer twice to determine the optimal qubit pair combination. The FC neural network, featuring two layers each with 1000 latent nodes, manages a fixed input size that matches the entire qubit set ($N_q^2 \times N_{\text{dim}}$) and outputs a decision for all possible qubit pairs, providing less flexibility for transfer learning compared to the Transformer. The main Transformer model, specifically designed for scheduling using qubit pair data, consists of 3 blocks but with an embedding dimension of 32 and a simpler structure, having a single head and a feed-forward network hidden dimension of 64. It aims to identify the next qubit pairs by analyzing the post-processed action matrix. The Transformer models are trained over 3000 epochs in the simulation environment, using a constant learning rate of $3 \times 10^{-3}$ with the Adam optimizer, and hyperparameters optimized through grid searching. The training process takes approximately two days on a single A30 GPU supported by a 24-core Intel Xeon E5 CPU.

The training process for the Transformer neural network begins with an initialization phase where the network is pre-trained to mimic the outputs of the Greedy-on-QuPair algorithm. This provides a baseline for the network's parameters. To introduce variability and enhance generalization, random variations are added to the network parameters. The training then proceeds iteratively, with the network updating its parameters based on the rewards obtained from Monte Carlo simulations. The goal of each update is to guide the network toward actions that maximize the reward. To improve scalability and training efficiency, the Transformer-on-Qupairs architecture leverages transfer learning. Specifically, the model trained for $N_q = 40$ qubits is used as the initial model for training the $N_q = 80$ model. Similarly, the $N_q = 80$ trained model serves as the starting point for training the $N_q = 120$ model. This progressive training approach significantly reduces the computational overhead and speeds up convergence for larger systems.

## 5.2 RESULTS

**Dynamic scheduling process** Figure 3 presents the quantum resource scheduling process for a system with $N_q = 40$ qubits. Figure 3a displays the progression of the simulation at various time steps ($N_t$). Figure 3b shows that as the entanglement trials advance, there is an increase in both the maximum cluster size and the systematic error, ultimately leading to the largest connected subgraph size $N_{\text{max}}$, which is further illustrated in Figure 3c. Interestingly, the logarithm of the system's quantum volume ($\mu = \log_2 V_Q$) reaches a peak at a specific $N_t$. This peak value of $\mu$ is used as our representative result in a Monte Carlo simulation. To establish a benchmark for different strategies, we conduct 100 simulations and calculate the average $\bar{\mu}$. The error bars represent a 2-sigma interval, used to compute the standard deviation $\sigma(\bar{\mu})$ across these samples.

**Scheduling methodological comparison** Table 1 presents a performance comparison of various scheduling methods measured by $\bar{\mu}$. We examine several rule-based strategies, including the random baseline, which is employed when detailed inhomogeneous information about the qubit resource is unavailable. The Static MST method is suitable for systems with long coherence times or high probabilities of successful heralded entanglement. In contrast, for more realistic quantum systems

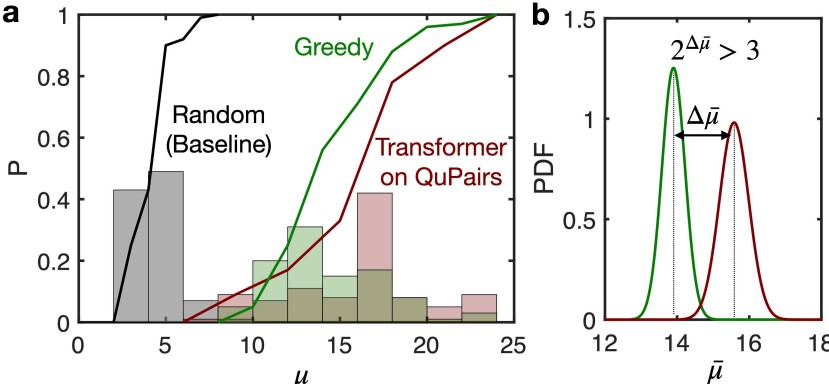

Figure 4: **Comparison of qubit cluster state building strategies. a,** Histograms of 100 samplings $\mu$ for various strategies: random (black), Greedy-on-QuPairs (green), and Transformer-on-QuPairs (red), with cumulative density functions overlaid. **b,** Probability density function (Gaussian fitting) comparison for $\bar{\mu}$ between Greedy and Transformer-on-QuPairs strategies, highlighting the superior optimization capacity of the Transformer. The $\Delta\bar{\mu}$ shows a benefit improvement in the quantum system, with $2^{\Delta\bar{\mu}} > 3$ indicating a significant enhancement.

with limited coherence times and lower entanglement success rates, a greedy heuristic that selects steps based on the minimum expected error performs best among the rule-based options.

In our examination of RL-based methods, the Transformer on individual qubit information, which does not incorporate qubit pair data, performs similarly to random scheduling. However, the fully-connected graph of the QuPairs, where output results are mixed with a 10% incorporation of the greedy action matrix predictions, is trainable and uses these predictions as a baseline, thereby achieving superior results compared to the standalone greedy method. The Transformer-on-QuPairs architecture, which inherently accounts for latent interactions between qubit pairs and supports scaling to larger input sequences, also mixes its outputs with the greedy predictions at a 10% ratio, and it outperforms the fully-connected architecture.

For comparative analysis, we focus on the most effective rule-based strategy, Greedy-on-QuPairs, and the best RL-based method, Transformer-on-QuPairs, to highlight the advantages of machine learning technologies. Given the intrinsic probabilistic nature of quantum state building, we conducted 100 Monte Carlo simulations to generate a histogram of the quantum volume distribution ($\mu = \log_2 V_Q$), illustrated in Figure 4a with three typical strategies: random (black baseline), Greedy-on-QuPairs (green), and Transformer-on-QuPairs (red). The analysis shows that the Greedy method, utilizing pre-characterized system information, significantly outperforms random sampling, underscoring the importance of inhomogenous info input. The Transformer strategy further enhances this by dynamically adjusting cost functions based on state changes, boosting system performance. This improvement could potentially be augmented through the use of Monte Carlo tree search techniques in resource scheduling. The probability density function (PDF) after Gaussian fitting for these two methods is plotted in Figure 4b, and the mean difference between these distributions is calculated by $\Delta\bar{\mu}$. Given that the quantum system's performance, evaluated by the quantum volume, exponentially improves with larger $\mu$, we estimate an enhancement of $2^{\Delta\bar{\mu}} > 3$ in quantum system benefits when transitioning from rule-based to RL-based scheduling with the Transformer-on-QuPairs architecture.

**Environmental variation comparison** Table 2 shows the performance distributions, highlighting the mean $\bar{\mu}$. Strategies that employ random scheduling, disregarding the inhomogeneity of the graph data, demonstrate the least effectiveness due to their non-optimized construction processes. Among various strategies, an increase in $\bar{\mu}$ is associated with a corresponding increase in $\sigma(\bar{\mu})$. Furthermore, environments with greater variability show a more pronounced advantage when utilizing the Transformer-on-QuPairs approach over the Greedy-on-QuPairs method.

**Qubit number scaling comparison** We also explore the scaling of qubits to augment the dynamic scheduling resource pool, summarized in Table 3. In our qubit scaling experiments, the agent

Table 2: **Environment variations sweep.** In a homogeneous qubit environment ($\sigma(F) = 0$), there is minimal variation in performance across different strategies. In contrast, more inhomogeneous environments (characterized by larger $\sigma(F)$) enhance the benefits of using the Transformer-on-QuPairs method, as indicated by $2^{\text{mean}(\Delta\bar{\mu})}$. Conversely, the performance of the random baseline strategy deteriorates as $\sigma(F)$ increases.

| $\sigma(F)$ | $\bar{\mu}$ - Random | $\bar{\mu}$ - Greedy-on-QuPairs | $\bar{\mu}$ - Transformer-on-QuPairs | $2^{\text{mean}(\Delta\bar{\mu})}$ |
|---|---|---|---|---|
| 0 | 4.61±0.16 | 4.63±0.16 | 4.65±0.16 | 1.01 |
| 0.03 | 4.50±0.17 | 12.66±0.61 | 13.09±0.66 | 1.35 |
| 0.06 | 4.18±0.18 | 13.74±0.66 | 14.93±0.76 | 2.28 |
| 0.09 | 3.85±0.23 | 13.90±0.62 | **15.58±0.84** | **3.20** |

Table 3: **Qubit number scaling comparison.** As the number of qubits ($N_q$) in the system increases, the Transformer-on-QuPairs strategies offer greater benefits compared to the rule-based Greedy methods, as reflected in the $2^{\text{mean}(\Delta\bar{\mu})}$ values. Meanwhile, the performance of the random baseline strategy declines with the scaling of $N_q$.

| $N_q$ | $\bar{\mu}$ - Random | $\bar{\mu}$ - Greedy-on-QuPairs | $\bar{\mu}$ - Transformer-on-QuPairs | $2^{\text{mean}(\Delta\bar{\mu})}$ |
|---|---|---|---|---|
| 40 | 3.85±0.23 | 13.90±0.62 | 15.58±0.84 | 3.20 |
| 80 | 3.38±0.18 | 14.11±0.70 | 15.85±0.88 | 3.34 |
| 120 | 3.18±0.17 | 14.20±0.92 | 16.12±0.95 | 3.78 |
| 160 | 2.97±0.15 | 14.32±1.06 | **16.51±1.21** | **4.56** |

designed for scheduling larger $N_q$ qubit resources takes advantage of a model trained on a smaller scale ($N_q = 40$). This transfer learning approach uses the same training configuration, running for 3000 epochs under identical conditions as those used for training from scratch. This strategy ensures consistent training settings while adapting the model to handle increased complexity due to more qubits. We observe that as the number of qubits increases, the performance of the random algorithm deteriorates, whereas both the greedy and Transformer-on-QuPairs methods demonstrate improved performance due to their enhanced flexibility in programmable operations. The Transformer-on-QuPairs method particularly shows greater benefits for quantum system performance with larger $N_q$ sizes. This improvement underscores the scalability of the Transformer-on-QuPairs method, facilitated by transfer learning settings, to effectively manage dynamic scheduling across varying qubit quantities for system optimization.

## 6 CONCLUSION

In this paper, the key achievements of this study include: (1), We formulate the quantum resource scheduling problem in a digitized environment with Monte Carlo Simulation. Such environment enables us to develop a rule-based greedy heuristic method that significantly outperforms the random scheduling baseline. (2), We train reinforcement learning agents in this interactive probabilistic environment with a Transformer-on-QuPairs framework that uses self-attention on the sequential information of inhomogeneous qubit pairs to provide the next-step dynamic scheduling guidance. Furthermore, this Transformer-on-QuPairs scheduler achieves a quantum system performance improvement of more than $3\times$ compared to our rule-based method in the inhomogeneous simulation experiment. (3), This framework is scalable, capable of handling larger sets of input variables like charge-state estimations and fidelity-rate trade-offs, that potentially extending beyond the current limit of 7 input dimensions. It also supports longer sequences to accommodate more interaction links as the number of qubits ($N_q$) increases. These advancements pave the way for new possibilities in the co-design of physical and control systems within the realms of quantum communication, networking, and computing.

## 7 BROADER IMPACTS AND DISCUSSION

Quantum computing has the potential to dramatically enhance fields like chemistry, notably in drug and material design, offering substantial societal benefits such as the development of new, effective pharmaceuticals. The increasing relevance of machine learning in quantum applications emphasizes the necessity for system-level optimization to manage control and scheduling tasks effectively across nonuniform qubit arrays. Our RL-based framework for system optimization is versatile, suitable for a range of large-scale quantum systems including superconducting qubits, artificial-atom quantum repeaters, neural atoms, and trapped ions. This research specifically utilizes RL to improve dynamic scheduling decisions, maximizing the effectiveness of existing hardware platforms. We foresee no negative impact from our research, no significant consequences from system failures, nor do we believe that our methods leverage any bias in any data. We did not perform any experiments on a real quantum machine. However, possible ethical and social impacts, such as the use for the development of chemical weapons, require careful scrutiny. This works also has limitations that using large sequence attention on qubit pairs ($N_q^2$) becomes computationally challenging for a too large number of $N_q$. Additionally, varying environments in the quantum resource distribution (different $F, R$ distributions) would require retraining of the reinforcement learning model to maintain optimal system performance.

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

## A  QUANTUM INFORMATION PROCESSING ADDITIONAL BACKGROUND

Since this paper is based on the quantum resource scheduling of cluster states, we introduce some of the basic concepts for the better clarity of the readers. We organize the appendix in the following ways. We start with the basic postulates in quantum mechanics and compare them with that in classical physics, as they are the building blocks for computing. Next, we talk about the fundamentals of quantum computing and compare it with its classical counterpart. Then we introduce the concept of quantum entanglement which is important to understand the notion of cluster states. We further discuss the basics of cluster states and their usage in computing, communication, resource scheduling. It is the resource scheduling part which is crucial for our paper. Physically, there are various ways to generate these cluster states, but for our paper we rely on atom-cavity system. We then introduce the formalism of atom-cavity-photodetection and since this is an open quantum system, we also talk about the equations of motion which describe its time evolution. We use quantum Monte-Carlo method to simulate these equations of motion which is further described. At last we talk about how to generate an entanglement using an atom-cavity system, one of the methods being the Barrett-Kok Protocol.

Readers can feel free to skip parts which they are already familiar with. Since quantum mechanics is a vast subject by itself, this appendix is just an attempt to very briefly summarize all the relevant concepts for an ease of understanding.

### A.1  POSTULATES IN QUANTUM MECHANICS (LANDAU & LIFSHITS, 1991; NIELSEN & CHUANG, 2010)

- **State Postulate**:

  Description: The state of a quantum system is fully described by a wavefunction $|\psi\rangle$ (for pure states) or a density matrix $\rho$ (for mixed states).

  Mathematical Forms: Pure state: $|\psi\rangle \in \mathcal{H}$, where $\mathcal{H}$ is a complex Hilbert space. Mixed state: $\rho = \sum_i p_i |\psi_i\rangle\langle\psi_i|$, where $p_i$ are probabilities and $|\psi_i\rangle$ are pure states, and $\langle\psi_i| \in \mathcal{H}^\dagger$ is the adjoint of the state $|\psi_i\rangle$.

- **Observable Postulate**:

  Description: Observables in quantum mechanics are represented by Hermitian (self-adjoint) operators $\hat{A}$ acting on the Hilbert space $\mathcal{H}$.

  Mathematical Forms: $\hat{A} = \hat{A}^\dagger$.

  Examples: The position operator $\hat{x}$ and the momentum operator $\hat{p}$.

- **Measurement Postulate**:

  Description: The measurement of an observable $\hat{A}$ yields one of its eigenvalues $a_n$ with a probability given by the Born rule.

  Mathematical Forms: The probability $P(a_n)$ of obtaining eigenvalue $a_n$ is $P(a_n) = |\langle\psi|\phi_n\rangle|^2$, where $|\phi_n\rangle$ is the eigenstate corresponding to $a_n$. After measurement, the system collapses to the eigenstate $|\phi_n\rangle$.

- **Time Evolution Postulate**:

  Description: The time evolution of a quantum state is governed by the Schrödinger equation.

  Mathematical Forms: For a pure state: $i\hbar\frac{\partial}{\partial t}|\psi(t)\rangle = \hat{H}|\psi(t)\rangle$, where $\hat{H}$ is the Hamiltonian. For a density matrix: $\frac{d\rho}{dt} = -\frac{i}{\hbar}[\hat{H}, \rho]$. Here, for any two operators $A$ and $B$, the operation $[A,B]$ is called the commutator given by $AB - BA$. Similarly $\{A,B\}$ is called the ant-commutator given by $AB + BA$.

- **Superposition Postulate**:

  Description: If a system can be in states $|\psi_1\rangle$ and $|\psi_2\rangle$, it can also be in any linear combination of these states.

  Mathematical Forms: $|\psi\rangle = c_1|\psi_1\rangle + c_2|\psi_2\rangle$, where $c_1$ and $c_2$ are complex coefficients.

- **Composite Systems Postulate**:

  Description: The state space of a composite quantum system is the tensor product of the state spaces of the individual subsystems.

Mathematical Forms: If systems A and B are described by $\mathcal{H}_A$ and $\mathcal{H}_B$, respectively, then the composite system is described by $\mathcal{H}_A \otimes \mathcal{H}_B$.

Example: For two particles $A$ and $B$, the combined state will be $|\psi\rangle = |\psi_A\rangle \otimes |\psi_B\rangle$.

## A.2 COMPARISON TO CLASSICAL MECHANICS (GOLDSTEIN, 1980)

- **State Description:** In classical mechanics, the state is described by precise values of position and momentum (phase space points), whereas in quantum mechanics, the state is described by a wavefunction or density matrix.

- **Observables and Measurement:** Classical observables have definite values and their measurement does not disturb the system. In contrast, quantum measurements generally disturb the system and the outcome is probabilistic.

- **Time Evolution:** Classical systems follow deterministic trajectories governed by Newton's laws or Hamilton's equations. Quantum systems evolve according to the Schrödinger equation, which is deterministic in terms of the wavefunction but probabilistic upon measurement.

- **Superposition:** Classical systems cannot exist in superpositions of states; they are always in a definite state. Quantum systems, however, can exist in superpositions, leading to phenomena like interference and entanglement.

## A.3 BASICS OF QUANTUM COMPUTING (NIELSEN & CHUANG, 2010)

- **Quantum Bits (Qubits):**

  Description: The basic unit of quantum information is the qubit, which can exist in a superposition of the basis states $|0\rangle$ and $|1\rangle$.

  Mathematical Form: A qubit state $|\psi\rangle$ is represented as $|\psi\rangle = \alpha|0\rangle + \beta|1\rangle$, where $\alpha$ and $\beta$ are complex numbers satisfying $|\alpha|^2 + |\beta|^2 = 1$.

  Physical Realization: A physical system having two quantum states can be encoded as a qubit. For example, the two least energetic states (ground $|g\rangle$ and excited $|e\rangle$) of an atom (Saffman, 2016), superconducting qubit (Blais et al., 2021), color center defects (Doherty et al., 2022), quantum dots (Brown et al., 2001), the two polarization states (horizontal $|H\rangle$ and vertical $|V\rangle$) of a photon (Wang et al., 2019), ion traps (Bruzewicz et al., 2019), the two time-bin (early $|E\rangle$ and late $|L\rangle$) of an incoming photon (Ward & Keller, 2022), can be each encoded as a qubit.

- **Quantum Gates:**

  Description: Quantum gates are the basic operations applied to qubits. They are represented by unitary matrices that manipulate qubit states through reversible transformations.

  Mathematical Form: For a single qubit, a quantum gate $U$ acts on the qubit state $|\psi\rangle$ as $U|\psi\rangle$. In the density matrix formalism, for the qubit state $\rho$, operation of $U$ transforms the state to $U\rho U^\dagger$

  Basic single qubit gates:

  Pauli-X Gate $\sigma_X = \begin{pmatrix} 0 & 1 \\ 1 & 0 \end{pmatrix}$ (similar to bit flip in classical computing)

  Pauli-Y Gate $\sigma_Y = \begin{pmatrix} 0 & -i \\ i & 0 \end{pmatrix}$ (similar to phase flip)

  Pauli-Z Gate $\sigma_Z = \begin{pmatrix} 1 & 0 \\ 0 & -1 \end{pmatrix}$

  Hadamard Gate $\frac{1}{\sqrt{2}} \begin{pmatrix} 1 & 1 \\ 1 & -1 \end{pmatrix}$

  Phase Gate $\begin{pmatrix} 1 & 0 \\ 0 & i \end{pmatrix}$

  $\pi/8$ Gate $\begin{pmatrix} 1 & 0 \\ 0 & e^{i\pi/4} \end{pmatrix}$

  Examples:

$\alpha|0\rangle + \beta|1\rangle \quad \beta|0\rangle + \alpha|1\rangle$

$\alpha|0\rangle + \beta|1\rangle \quad \alpha|0\rangle - \beta|1\rangle$

$\alpha|0\rangle + \beta|1\rangle \quad \alpha\frac{|0\rangle+|1\rangle}{\sqrt{2}} + \beta\frac{|0\rangle-|1\rangle}{\sqrt{2}}$

Basic two qubit gate:

CNOT Gate: $\begin{pmatrix} 1 & 0 & 0 & 0 \\ 0 & 1 & 0 & 0 \\ 0 & 0 & 0 & 1 \\ 0 & 0 & 1 & 0 \end{pmatrix}$

Universal Gate set: Set of gates from which any quantum operation can be constructed. Such a set allows for the implementation of any quantum algorithm. For example, any multiple qubit gate can be represented as the composition of a CNOT and single qubit gates.

Physical Realization: To implement a quantum gate, usually laser (optical) (Greilich et al., 2009), radio-frequency (Bardin et al., 2021) (RF/microwave) electromagnetic field, or mechanical wave (Hong et al., 2012) is used, which probes the energy levels of the quantum system.

- **Quantum Algorithms:**

  Description: Quantum algorithms leverage quantum superposition, entanglement, and interference to solve certain problems more efficiently than classical algorithms.

  Examples:

  Shor's Algorithm (Shor, 1997): Efficiently factorizes large integers, exponentially faster than the best-known classical algorithms.

  Grover's Algorithm (Grover, 1996): Searches an unsorted database of $N$ items in $O(\sqrt{N})$ time, providing a quadratic speedup over classical algorithms.

## A.4 COMPARISON TO CLASSICAL COMPUTING

- **Bits vs. Qubits**: Classical bits can be either 0 or 1, while qubits can exist in superpositions of 0 and 1, allowing quantum computers to process a vast amount of information simultaneously. Bits are physically realized using transistors, whereas qubits rely on quantum states of different physical systems (superconducting qubit, ion-traps, atom-arrays, color-centers, quantum dots, photons etc.)

- **Deterministic vs. Probabilistic**: Classical gates perform deterministic operations on bits. Quantum gates perform unitary transformations, leading to probabilistic outcomes upon measurement.

- **No Entanglement**: Classical bits are independent, while qubits can be entangled, creating correlations that enable more powerful computational techniques.

## A.5 INTRODUCTION TO QUANTUM ENTANGLEMENT (NIELSEN & CHUANG, 2010)

- **Definition**:

  Description: Entanglement occurs when the quantum state of a composite system cannot be factored into states of individual subsystems.

  Mathematical Form: For a system with two qubits $A$ and $B$, an entangled state $|\psi_{AB}\rangle$ cannot be expressed as $|\psi_A\rangle \otimes |\psi_B\rangle$. For example, $|\psi\rangle = \frac{1}{\sqrt{2}}(|00\rangle + |11\rangle)$ is an entangled state.

- **Measurement of Entangled Particles**:

  Description: Measurement of one particle in an entangled pair instantaneously determines the state of the other particle due to their correlated nature.

  Mathematical Form: For the entangled state $|\psi\rangle = \frac{1}{\sqrt{2}}(|00\rangle + |11\rangle)$, measuring qubit $A$ in state $|0\rangle$ collapses the entire state to $|00\rangle$, and measuring qubit $A$ in state $|1\rangle$ collapses it to $|11\rangle$.

- **Bell States (Maximally Entangled States)**:

  Description: Bell states represent the simplest and most well-known examples of maximally entangled states.

Mathematical Form:

The four Bell states for two qubits are:

$|\Phi^+\rangle = \frac{1}{\sqrt{2}}(|00\rangle + |11\rangle)$; $|\Phi^-\rangle = \frac{1}{\sqrt{2}}(|00\rangle - |11\rangle)$; $|\Psi^+\rangle = \frac{1}{\sqrt{2}}(|01\rangle + |10\rangle)$; $|\Psi^-\rangle = \frac{1}{\sqrt{2}}(|01\rangle - |10\rangle)$

- **Entanglement as a resource**: Entanglement is used as a resource for various quantum protocols in the field of quantum networks, quantum communication and quantum computing. There exist schemes based on quantum repeaters which extend entanglement between distant nodes. For this paper, we particularly focus on cluster states as a resource.

## A.6 USING CLUSTER STATES AS A RESOURCE (NIELSEN, 2006)

- **Definition of Cluster States**:

  Description: A cluster state is a type of multi-qubit entangled state that can be used as a universal resource for quantum computation through a series of adaptive measurements.
  Mathematical Form:
  1D cluster state of $N$ qubits:

$$|\phi_N\rangle = \frac{1}{2^{N/2}} \otimes_{a=1}^{N} (|0\rangle_a Z^{a+1} + |1\rangle_a) \tag{1}$$

GHZ state:

$$|GHZ_N\rangle = \frac{1}{\sqrt{2}}(|0\rangle_1|0\rangle_2...|0\rangle_N + |1\rangle_1|1\rangle_2...|1\rangle_N) \tag{2}$$

W state:

$$|W_N\rangle = \frac{1}{\sqrt{N}}(|1\rangle_1|0\rangle_2...|0\rangle_N + |0\rangle_1|1\rangle_2...|0\rangle_N + ... + |0\rangle_1|0\rangle_2...|0\rangle_{N-1}|1\rangle_N) \tag{3}$$

- **Universal Quantum Computation (DiVincenzo et al., 2000)**: Any quantum algorithm can be implemented on a cluster state through a sequence of single-qubit measurements and classical feedforward. Logical quantum gates are implemented by performing measurements on the qubits in the cluster state.

- **Entanglement Distribution (Inagaki et al., 2013)**: Cluster states can be used to distribute entanglement across nodes in a quantum network for various protocols.

- **Quantum Communication (Gisin & Thew, 2007)**: Cluster states enable protocols like quantum teleportation and superdense coding over a network.

- **Error Correction and Fault Tolerance Egan et al. (2021)**: Use redundancy and entanglement in cluster states to detect and correct errors through syndrome measurements and classical processing.

- **Resource Sharing (Qian et al., 2015)**: Cluster states enable the sharing of quantum resources (e.g., entanglement) across different parts of a quantum network. Different parts of a cluster state can be used for different tasks such as entanglement swapping, teleportation, and secure communication.

- **Generating Cluster States (Rohde & Barrett, 2007)**: Cluster states are built upon pair wise entanglement. For this paper, we particularly focus on generating entanglement using spin-photon interfaces and beam splitter. An example of a spin-photon interface is an atom-cavity system. Here, atomic system has the spin qubit whereas the cavity has the photonic qubit. Since the atom and cavity is coupled, a spin-photon interface generates entanglement between the spin qubit and the photonic qubit.

  If we consider two spin-photon interfaces A and B, the photons generated by them are passed through a beam splitter. When photons from two sources pass through the beam splitter, they interfere and leads to clicks in the detectors. Heralding on the clicks is equivalent to projecting the photonic qubits on a Bell-basis. Since these photonic qubits are entangled to the spins A and B individually, projecting the photonic qubits into Bell basis, leaves a residual entanglement between spin A and B.

### A.7 FORMALISM OF ATOM-CAVITY INTERACTION (JANITZ ET AL., 2020)

- **Atom-Cavity Systems**: The interaction of a two-level atom with a single mode of a quantized electromagnetic field in a cavity is described by the Jaynes-Cummings Hamiltonian:

$$\hat{H}_{JC} = \frac{\hbar\omega_0}{2}\hat{\sigma}_z + \hbar\omega\hat{a}^\dagger\hat{a} + \hbar g(\hat{\sigma}_+\hat{a} + \hat{\sigma}_-\hat{a}^\dagger) \qquad (4)$$

  where $\omega_0$ is the transition frequency of the two-level atom, $\omega$ is the frequency of the cavity mode, $g$ is the coupling strength between the atom and the cavity mode, $\hat{\sigma}_z$ is the Pauli z-operator for the two-level atom, $\hat{\sigma}_+$ and $\hat{\sigma}_-$ are the raising and lowering operators for the atomic states, respectively, $\hat{a}^\dagger$ and $\hat{a}$ are the creation and annihilation operators for the cavity photons. This formalism has no direct classical counterpart but can be seen as analogous to a classical resonator coupled to a harmonic oscillator.

- **Rabi Oscillations**: These are coherent oscillations in the probability amplitude of the atomic states due to their interaction with the cavity mode.

- **Purcell Enhancement:** The interaction of an atom with a cavity can significantly modify the spontaneous emission properties of the atom, a phenomenon known as the Purcell effect.
  Purcell Factor: The enhancement of the spontaneous emission rate $\Gamma$ in the presence of a cavity is quantified by the Purcell factor $F_P$:

$$\Gamma_{cav}/\Gamma_{free} = F_P = \frac{3}{4\pi^2}\left(\frac{\lambda}{n}\right)^3\frac{Q}{V} \qquad (5)$$

  where $\Gamma_{cav}$ and $\Gamma_{free}$ is the spontaneous emission rate of the atom in cavity and free space respectively, $\lambda$ is the vacuum wavelength of the emitted light, $n$ is the refractive index of the medium, $Q$ is the quality factor of the cavity, which measures the sharpness of the resonance, $V$ is the mode volume of the cavity.
  Physical Interpretation: The Purcell factor indicates how much the emission rate is enhanced due to the cavity. A high $Q/V$ ratio means that the cavity strongly enhances the interaction between the atom and the electromagnetic field, increasing the emission rate. This is particularly useful in applications like single-photon sources and quantum information processing.

### A.8 LINDBLAD EQUATION OF MOTION FOR OPEN QUANTUM SYSTEMS (SCULLY & ZUBAIRY, 1997)

- **Open Quantum Systems**: Quantum systems interacting with their environment are described by the Lindblad master equation, which includes both unitary and dissipative dynamics. The dissipative dynamics is also referred to as the process of decoherence.
  To have an operational and useful qubit, one mainly requires three counter-acting abilities:
  (1) **Control qubit:** to efficiently control (initialize, manipulate, readout) the qubit
  (2) **Memory qubit:** to store quantum information in the qubit
  (3) **Communication qubit:** to transmit quantum information between multiple locations
  (1) requires qubit to have controlled interaction with the environment which also leads to some amount of decoherence, whereas (2) demands qubits to be completely isolated from the environment to increase the storage time (also known as T1 and T2). On the other hand, (3) requires to store quantum information in the type of qubits which can travel fast between different spatial locations.
  In order to resolve each of these points in a single system, we use a spin-photon interface with the following set of qubits:
  (1) **Electron-spin qubit**: color-center defects have electronic-energy level structure which closely resembles to that of a two level structure, and is efficiently controllable by microwave B-field with high fidelity. Hence, they are ideal as a control qubit, but they suffer decoherence due to their coupling with the environment.
  (2) **Nuclear spin qubit**: defect centers also have nuclear spins which are isolated from the environment and hence have low decoherence. Using hyperfine coupling, the electron spin qubit and nuclear spin qubit can interact and exchange quantum information. This makes nuclear spins ideal for quantum memory purposes.

(3) **Photon qubit**: implanting a color center qubit in an optical cavity, couples the electron spin qubit with a photon qubit, realizing a spin-photon interface.

This architecture, gives the ability to exchange quantum information between control, memory and communication qubits and use each to the best of their abilities.

- **Lindblad Master Equation**: The Lindblad equation for the density matrix $\rho$ is:

$$\frac{d\rho}{dt} = -\frac{i}{\hbar}[H, \rho] + \sum_k \left( L_k \rho L_k^\dagger - \frac{1}{2}\{L_k^\dagger L_k, \rho\} \right) \quad (6)$$

Here, $L_k$ are the Lindblad operators representing various environmental interactions, such as decay or dephasing.

- **Interpretation**: The first term $-\frac{i}{\hbar}[H, \rho]$ represents the coherent evolution, while the second term accounts for the dissipative processes due to the environment.

- **Example**: For spontaneous emission in a two-level atom, the Lindblad operator is $L = \sqrt{\gamma}\hat{\sigma}_-$, where $\gamma$ is the decay rate and $\hat{\sigma}_-$ is the lowering operator.

## A.9 Quantum Monte-Carlo/Jump Method (Mølmer et al., 1993)

- **Quantum Jump Method:** Density matrix formalism deals with the ensemble average over multiple realizations of the system evolution, whereas the Quantum Jump (Monte-Carlo) method allows to simulate the system dynamics individually. Environment is continuously monitored in the form of quantum measurements (in our case detecting photon clicks in detector), and each measurement leads to *collapsing* of wavefunction to a pure state.

- **Non-Hermitian Evolution:** The evolution above is described by Schrödinger equation with a non-Hermitian effective Hamiltonian:

$$H_{eff} = H_{sys} - \frac{i\hbar}{2}\sum_n C_n^\dagger C_n \quad (7)$$

where $C_n$ are the collapse operators each corresponding to irreversible processes present in the system, with rate $\gamma_n$. Due to the non-Hermitian nature of the Hamiltonian, the norm of the wavefunction reduces in a small time $\delta t$, given by $\langle\psi(t+\delta t)|\psi(t+\delta t)\rangle = 1 - \delta p$, where

$$\delta p = \delta t \sum_n \langle\psi(t)|C_n^\dagger C_n|\psi(t)\rangle \quad (8)$$

- **Collapsing of the Wavefunction:** If there is a quantum jump registered by environmental measurements, for example photon emitted by the atom-cavity system being detected by the photodetector, it leads to a quantum jump to the state $|\psi(t+\delta t)\rangle$, obtained by projecting the previous state $|\psi(t)\rangle$ via the collapse operator $C_n$ corresponding to the measurement:

$$|\psi(t+\delta t)\rangle = \frac{C_n|\psi(t)\rangle}{\langle\psi(t)|C_n^\dagger C_n|\psi(t)\rangle^{1/2}} \quad (9)$$

Similarly, the probability of collapse due to the $i^{th}$ collapse operator $C_i$ is given by:

$$P_i(t) = \frac{\langle\psi(t)|C_i^\dagger C_i|\psi(t)\rangle}{\delta p} \quad (10)$$

- **QuTiP algorithm (qut, 2011):** Quantum Monte Carlo evolution is tedious, therefore QuTiP uses the following algorithm to simulate the system:

**1. Initialization:** Start from an initial pure state $|\psi(0)\rangle$.

**2. Random number selection:** Choose a random number $r$ between 0 and 1, which would represent the probability that a quantum jump occurs.

**3. Integration:** Integrate the Schrödinger equation using the effective Hamiltonian $H_{eff}$ in Eq. 7 until a time $\tau$, such that $\langle\psi(\tau)|\psi(\tau)\rangle = r$, at which point a quantum jump occurs.

**4. Selecting the collapse operator:** Select the collapse operator $C_k$ such that $k$ is the smallest integer which satisfies:

$$\sum_{i=1}^k P_i(\tau) \geq r. \quad (11)$$

**5. Renormalization:** After projecting the state using the collapse operator $C_k$, obtain the new renormalized state using Eq. 9.

**6. Repeat:** Using the state obtained in step (5) as an initial state, repeat from step (1), unless the simulation time is reached.

### A.10 BARRETT-KOK (BK) PROTOCOL OVERVIEW (BARRETT & KOK, 2005)

- **Purpose**: The Barrett-Kok protocol aims to generate high-fidelity entangled states between distant qubits, crucial for quantum communication and distributed quantum computing. Fig. 6(a) shows the experimental layout for implementing the BK protocol. Fig. 6(b) shows the energy-level diagram of the electron spin qubit in a group-IV defect center in diamond.

- **Entanglement Generation**: The protocol typically involves using atom-cavity systems and beam-splitter to mediate entanglement between remote qubits. Readers are suggested to simultaneously refer to Fig. 6 to understand the experimental implementation of the following steps. The steps include:

  **1. Initialization**: The electron spin qubit is initialized to the state $\frac{|g\downarrow\rangle + |g\uparrow\rangle}{2}$ using laser and microwave antenna. Proceed to step 2.

  **2. Interaction**: System A(B) is equivalent to an atom-cavity system, therefore the Hamiltonian $H_0$ in Eq. 12 and Fig. 7 is similar in form to the Jaynes-Cummings Hamiltonian in Eq. 4. Proceed to step 3.

  **3. System Evolution**: Both the system evolves w.r.t the Hamiltonian described in step 2. The evolution leads to creation of photon qubits which pass through optical fibres due to its coupling with the optical cavity. Proceed to step 4.

  **4. Beamsplitter**: The photons emitted by two atom-cavity system passes through the two input ports of the 50:50 beamsplitter. Beam-splitter erases the which-path information of the incoming photons due to interference. Proceed to step 5.

  **5. Monitoring**: Start the round by waiting for upto time $t_{wait}$ for a photo-detection event in detectors $D_1$ and $D_2$. Monitor the clicks on the detectors. If this is first round and there is no click, the protocol fails and re-start from step 1. If this is first round and if there is a single photo-detection event in this round, wait further for time $t_{relax}$ for the remaining excitation in the system to relax, then proceed to step 6. If this is second round and there is no click, the protocol fails and re-start from step 1. If this is second round and if there is a single photo-detection event in this round, wait further for time $t_{relax}$ for the remaining excitation in the system to relax, then proceed to step 7.

  **6. Conditional Operations**: Apply an X gate (Appendix A.3) on the electron spin qubit using a MW antenna. Go back to step 3.

  **7. Additional swap**: Reaching this step means a successful entanglement attempt between electron spin qubit A and B. An additional step which is not included in the original BK scheme, but something which we propose in our protocol is the step of entanglement swapping, which means implementing electron-nuclear quantum SWAP gates via MW antenna, to swap the state from electron spin qubit to nuclear spin qubit, which serves as a quantum memory. Thus, by swap gates the entanglement between spin qubits is transferred to that between the nuclear spin qubits at two distant nodes A and B.

## B QMCS SIMULATION

The quantum simulation of the Barrett-Kok protocol starts by specifying the parameters for an atom within a cavity. A comprehensive detail of the entire protocol is presented in Appendix A.10, Fig. 5, 6a, 7 and in Alg. 1. The primary Hamiltonian for a dual atom-cavity system, relative to a frame

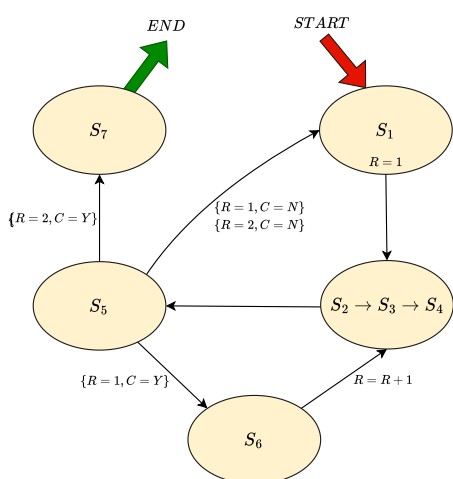

Figure 5: **Finite State Machine (FSM) representation of the BK protocol.** This loop from START to END corresponds to a single succesful entanglement attempt of the protocol. Here $S_i$ represents the steps mentioned in Appendix A.10. The variable $R$ corresponds to the round number, and $C = N$ means *no click*, $C = Y$ means *single photon click*.

rotating at $\omega_0$, is expressed as follows (Appendix A.7):

$$
\begin{aligned}
\hat{H}_0 = {} & \hbar\Delta\omega_{c,A}\hat{a}^\dagger\hat{a} + \hbar\Delta\omega_{c,B}\hat{b}^\dagger\hat{b} + \frac{\hbar\Delta\omega_{\downarrow,A}}{2}\hat{\sigma}_{z\downarrow,A} \\
& + \frac{\hbar\Delta\omega_{\uparrow,A}}{2}\hat{\sigma}_{z\uparrow,A} + \frac{\hbar\Delta\omega_{\downarrow,B}}{2}\hat{\sigma}_{z\downarrow,B} + \frac{\hbar\Delta\omega_{\uparrow,B}}{2}\hat{\sigma}_{z\uparrow,A} \\
& - g_A(\hat{\sigma}^+_{\downarrow,A}\hat{a} + \hat{\sigma}^-_{\downarrow,A}\hat{a}^\dagger + \hat{\sigma}^+_{\uparrow,A}\hat{a} + \hat{\sigma}^-_{\uparrow,A}\hat{a}^\dagger) \\
& - g_B(\hat{\sigma}^+_{\downarrow,B}\hat{b} + \hat{\sigma}^-_{\downarrow,B}\hat{b}^\dagger + \hat{\sigma}^+_{\uparrow,B}\hat{b} + \hat{\sigma}^-_{\uparrow,B}\hat{b}^\dagger) \\
& + \hbar\Omega_A(t)(\hat{\sigma}^+_{\downarrow,A} + \hat{\sigma}^-_{\downarrow,A} + \hat{\sigma}^+_{\uparrow,A} + \hat{\sigma}^-_{\uparrow,A}) \\
& + \hbar\Omega_B(t)(\hat{\sigma}^+_{\downarrow,B} + \hat{\sigma}^-_{\downarrow,B} + \hat{\sigma}^+_{\uparrow,B} + \hat{\sigma}^-_{\uparrow,B}).
\end{aligned}
\tag{12}
$$

The atom is modeled as a 4-level system $\{|g\downarrow\rangle, |g\uparrow\rangle, |u\downarrow\rangle, |u\uparrow\rangle\}$ as seen in Fig. 6b, where we use the following nomenclatures for the operators and coefficients:

$$\hat{\sigma}_{z\downarrow} \equiv (|u\downarrow\rangle\langle u\downarrow| - |g\downarrow\rangle\langle g\downarrow|) \tag{13a}$$

$$\hat{\sigma}_{z\uparrow} \equiv (|u\uparrow\rangle\langle u\uparrow| - |g\uparrow\rangle\langle g\uparrow|) \tag{13b}$$

$$\hat{\sigma}^+_{\downarrow} \equiv |u\downarrow\rangle\langle g\downarrow| \tag{13c}$$

$$\hat{\sigma}^-_{\downarrow} \equiv |g\downarrow\rangle\langle u\downarrow| \tag{13d}$$

$$\hat{\sigma}^+_{\uparrow} \equiv |u\uparrow\rangle\langle g\uparrow| \tag{13e}$$

$$\hat{\sigma}^-_{\uparrow} \equiv |g\uparrow\rangle\langle u\uparrow| \tag{13f}$$

$$\hat{\sigma}^-_{\downarrow\uparrow} \equiv |g\uparrow\rangle\langle u\downarrow| \tag{13g}$$

$$\hat{\sigma}^-_{\uparrow\downarrow} \equiv |u\uparrow\rangle\langle g\downarrow| \tag{13h}$$

$$\Delta\omega_c = \omega_{cav} - \omega_0 \tag{13i}$$

$$\Delta\omega_\downarrow = \omega_\downarrow - \omega_0 \tag{13j}$$

$$\Delta\omega_\uparrow = \omega_\uparrow - \omega_0 \tag{13k}$$

where labels A and B correspond to system A and B, $\hat{a}$ ($\hat{a}^\dagger$) and $\hat{b}$ ($\hat{b}^\dagger$) correspond to the annihilation (creation) operator for cavity A and B respectively, $\omega_{cav}$ is the cavity frequency, $\omega_\downarrow$ is the frequency of the transition $|g\downarrow\rangle \leftrightarrow |u\downarrow\rangle$, $\omega_\uparrow$ is the frequency of the transition $|g\uparrow\rangle \leftrightarrow |u\uparrow\rangle$, $g_A$ and $g_B$ is the atom-cavity coupling strength for system A and B respectively, $\Omega_A(t)$ and $\Omega_B(t)$ is the gaussian laser-driving strength for atom A and B respectively. The pulse envelope $\Omega_{A(B)}(t)$ is selected so that the laser drive performs a perfect optical $\sigma_X$-gate in the basis $\{|g\rangle, |u\rangle\}$. To incorporate losses into the system, we use the Lindblad equation of motion for the density matrix and the Lindblad superoperators $\gamma_i \mathscr{L}_i$ (Appendix A.8):

$$\frac{d}{dt}\rho = \frac{1}{i\hbar}\left[\hat{H}_0, \rho\right] + \sum_i \gamma_i \mathscr{L}_i(\rho) \tag{14}$$

where

$$\gamma_i \mathscr{L}_i(\rho) = \frac{\gamma_i}{2}\left(2\hat{c}_i \rho \hat{c}_i^\dagger - \left\{\hat{c}_i^\dagger \hat{c}_i, \rho\right\}\right) \tag{15}$$

$$\hat{c}_i \in \left\{\hat{\sigma}_{\downarrow,A}^-, \hat{\sigma}_{\downarrow,B}^-, \hat{\sigma}_{\uparrow,A}^-, \hat{\sigma}_{\uparrow,B}^-, \hat{\sigma}_{\downarrow\uparrow,A}^-, \hat{\sigma}_{\downarrow\uparrow,B}^-, \hat{\sigma}_{\uparrow\downarrow,A}^-, \hat{\sigma}_{\uparrow\downarrow,B}^-, \hat{\sigma}_{z\downarrow,A}, \hat{\sigma}_{z\downarrow,B}, \hat{\sigma}_{z\uparrow,A}, \hat{\sigma}_{z\uparrow,B}, \hat{a}, \hat{b}, \right.$$
$$\left. \frac{\hat{a}+\hat{b}}{\sqrt{2}}, \frac{\hat{a}-\hat{b}}{\sqrt{2}} \right\} \tag{16}$$

$$\gamma_i \in \left\{\gamma_A, \gamma_B, \gamma_A, \gamma_B, \frac{\gamma_A}{\chi_A}, \frac{\gamma_B}{\chi_B}, \frac{\gamma_A}{\chi_A}, \frac{\gamma_B}{\chi_B}, K_A^{dep}, K_B^{dep}, K_A^{dep}, K_B^{dep}, \kappa_A, \kappa_B, K_A^{det}, K_B^{det}\right\}, \tag{17}$$

where $\gamma_{A(B)}$ corresponds to the spontaneous decay rate of atom A(B), $\chi_{A(B)}$ corresponds to the cyclicity of the spin-conserving transitions of atom A(B), $\kappa_{A(B)} + K_{A(B)}^{det}$ corresponds to the decay rate of cavity A(B), $K_{A(B)}^{dep}$ corresponds to the optical-dephasing rate of atom A(B), $K_{A(B)}^{det}$ corresponds to the coupling rate to detector A(B). We assume that $\Delta\omega_\uparrow >> \Delta\omega_\downarrow$, because in this limit the coupling of the optical transition $|g\uparrow\rangle \leftrightarrow |u\uparrow\rangle$ to the laser and cavity can be ignored as it is highly detuned, which simplifies the QMC simulation and reduces the run-time. The collapse operators of interest are:

$$\hat{c}_A = \frac{\hat{a}+\hat{b}}{\sqrt{2}}, \ \hat{c}_B = \frac{\hat{a}-\hat{b}}{\sqrt{2}} \tag{18}$$

The occurrence of collapse operators $\hat{c}_A$ and $\hat{c}_B$ corresponds to getting a click in detectors A and B, respectively, which is important information as seen in Fig. 6. Given a set of simulation parameters, we can run a quantum Monte Carlo solver (Appendix A.9) with $n_{\text{traj}}$ number of trajectories for the Hamiltonian $\hat{H}_0$. The solution leads to multiple quantum trajectories, which we divide based on whether the operators $\hat{c}_A$ or $\hat{c}_B$ occurred or not. For trajectories with a click on detectors A or B, we perform a conditional microwave (MW) Hamiltonian $\hat{H}_\pi$ on the final state, given by:

$$\hat{H}_\pi = \hbar\Omega_{MW}(t)(\hat{\sigma}_{MW,A}^+ + \hat{\sigma}_{MW,A}^-)$$
$$+ \hbar\Omega_{MW}(t)(\hat{\sigma}_{MW,B}^+ + \hat{\sigma}_{MW,B}^-), \tag{19}$$

where $\Omega_{MW}(t)$ is microwave gaussian driving strength for the transitions $|g\downarrow\rangle \leftrightarrow |g\uparrow\rangle$, and $\hat{\sigma}_{MW,A}^+$ ($\hat{\sigma}_{MW,A}^-$) and $\hat{\sigma}_{MW,B}^+$ ($\hat{\sigma}_{MW,B}^-$) corresponds to the raising: $|g\uparrow\rangle\langle g\downarrow|$ (lowering: $|g\downarrow\rangle\langle g\uparrow|$) operator for the MW transitions of atom A and B respectively. The pulse envelope $\Omega_{MW}(t)$ is selected so that $\hat{H}_\pi$ performs a perfect $\sigma_X$-gate in the basis $\{|g\downarrow\rangle, |g\uparrow\rangle\}$. After applying $\hat{H}_\pi$, we apply $\hat{H}_0$ again. This again leads to three possibilities of getting clicks on detectors A, B, or no clicks.

In this scheme, we applied $\hat{H}_0$ twice and, of all quantum trajectories, we classify only those trajectories as *good* which leads to a click on detector A or B after each application of $\hat{H}_0$. The trajectory picture can be seen in the Fig. 7. We can divide the good trajectories into 4 types: {AA, AB, BA, BB}. AA means detector A clicks both times, AB means detector A clicks first and detector B clicks second, BA means detector B clicks first and detector A clicks second, and BB means detector B clicks both times. QMCS gives the statistics for each of these trajectory types: $\{n_{AA}, n_{AB}, n_{BA}, n_{BB}\}$. For each of the good trajectories, we take the partial trace over the photon degrees of freedom and average over the density matrix for the 4 different types of *good* trajectories giving: $\{\rho_{AA}, \rho_{AB}, \rho_{BA}, \rho_{BB}\}$.

Using these values, we obtain the following expression for the fidelity of the Bell pair $F$ and the success probability $R$:

$$
\begin{aligned}
F_1 &= F(\rho_{AA}, \Phi_+), R_1 = n_{AA}\mathcal{N}/n_{\text{traj}}^2 \\
F_2 &= F(\rho_{AB}, \Phi_-), R_2 = n_{AB}\mathcal{N}/n_{\text{traj}}^2 \\
F_3 &= F(\rho_{BA}, \Phi_-), R_3 = n_{BA}\mathcal{N}/n_{\text{traj}}^2 \\
F_4 &= F(\rho_{BB}, \Phi_-), R_4 = n_{BB}\mathcal{N}/n_{\text{traj}}^2,
\end{aligned}
\tag{20}
$$

where $\Phi_{+(-)}$ are the Bell states given by (Appendix A.5):

$$
\Phi_{+(-)} = \frac{|\uparrow\downarrow\rangle_{AB} \pm |\downarrow\uparrow\rangle_{AB}}{\sqrt{2}},
\tag{21}
$$

and $F$ is the fidelity function such that for any two density matrices $\rho_1$ and $\rho_2$ we have:

$$
F(\rho_1, \rho_2) = \text{Tr}\left(\sqrt{\sqrt{\rho_1}\rho_2\sqrt{\rho_1}}\right),
\tag{22}
$$

and $n_{\text{traj}}$ is the number of trajectories for which the QMCS runs for, and $\mathcal{N}$ is a normalization constant. Using this, we evaluate the cost function as follows:

$$
C = \min_i(1 - F_i e^{-1/r_{\text{ent}}T_{\text{mem}}R_i}).
\tag{23}
$$

One of the four branches ($i = 1, 2, 3, 4$) is selected which minimizes the cost function above, to report the fidelity ($F_i$) and success rate ($r_{\text{ent}}R_i$) of the established entanglement between the qubit pair.

**Algorithm 1** QMCS BK Protocol

---

Initialize system and simulation parameter set $\{p\}$
Define spin-photon operators for systems A and B
Define Hamiltonians $\hat{H}_0$ and $\hat{H}_\pi$
Define the collapse operator set with $\hat{c}_A$ and $\hat{c}_B$

   **function** COST-FUNCTION($\{x\}$)
       Initialize $\rho$ and $counts$
       Run mcsolver for $\hat{H}_0$ using $\{x\}$ and $\{p\}$
       **for** (i $\leq n_{\text{traj}}$):
         **if** ($\hat{c}_A$ happened):
           Run mesolver for $\hat{H}_\pi$
           Run mcsolver for $\hat{H}_0$
           **for** (j $\leq n_{\text{traj}}$):
             **if** ($\hat{c}_A$ happened):
               Append final state to $\rho[0]$
               Increment $counts[0]$
             **if** ($\hat{c}_B$ happened):
               Append final state to $\rho[1]$
               Increment $counts[1]$
         **if** ($\hat{c}_B$ happened):
           Run mesolver for $\hat{H}_\pi$
           Run mcsolver for $\hat{H}_0$
           **for** (k $\leq n_{\text{traj}}$):
             **if** ($\hat{c}_A$ happened):
               Append final state to $\rho[2]$
               Increment $counts[2]$
             **if** ($\hat{c}_B$ happened):
               Append final state to $\rho[3]$
               Increment $counts[3]$
       Evaluate $C$ using Eq. 23
       **return** $C$
   **end function**

---

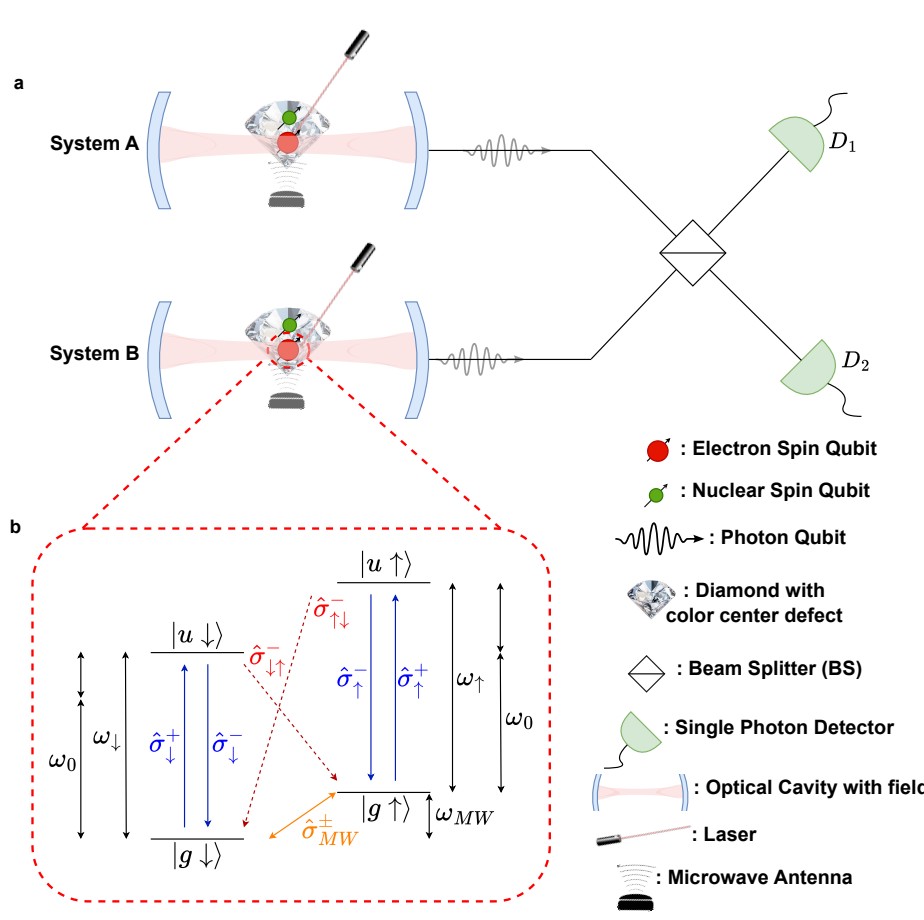

Figure 6: **Protocol layout example. a,** System A (B) comprises of an optical cavity with atomic qubit (diamond based color center defect as example). This atomic qubit has an electron spin qubit and a nuclear spin qubit. Laser is used to initialize and readout the electron spin qubit. The microwave antenna is used to implement quantum gates on the electron and nuclear spin qubit. The optical cavity is coupled to optical fibers allowing the transmission of the leaked photon qubit. The two photon qubits pass through the input ports of the beam splitter. The detectors $D_1$ and $D_2$ connected to the output ports of the beam splitter are monitored for a photon click. **b,** Four-level atomic system illustration for a atomic qubit (diamond based color center defect as an example). In color show the operators, blue, red, and orange representing the spin-conserving, spin-flipping, and MW transitions respectively.

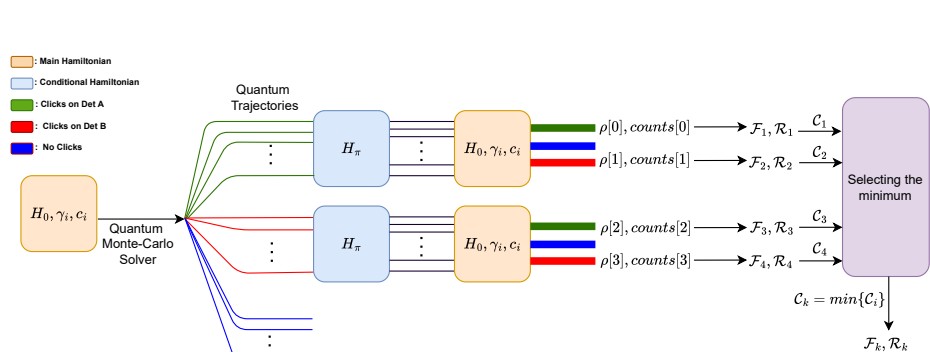

Figure 7: **QMCS visualization.** Barrett-Kok protocol visualization.

