# OpenReview forum: "Dynamic Inhomogeneous Quantum Resource Scheduling with Reinforcement Learning"
_ICLR.cc/2025/Conference — Submitted to ICLR 2025_

### Official Review · Reviewer_tMcL · 2024-10-24

**Soundness:** 2
**Presentation:** 2
**Contribution:** 3
**Rating:** 8
**Confidence:** 2

**Summary:**

This paper addresses the challenge of optimising the construction of entangled resource states in networked quantum computing architectures, where quantum bits are probabilistically entangled with one another. This is an extremely challenging problem, as the optimal control and scheduling of quantum resources is affected by the inhomogeneous nature of the underlying quantum resources, their pairwise interactions and probabilistic outcomes of remote entanglement. Optimising quantum resource state production is an NP-hard problem, which the authors tackle by simulating quantum resource scheduling in a digitised environment. Connections are made to the Minimum Weight Connected Subgraph Problem and the authors introduce a novel approach using reinforcement learning agents and a Transformer model, leveraging self-attention mechanisms to optimise qubit pair scheduling. This method enhances entangled resource state construction, yielding over a 3× improvement compared to traditional rule-based agents, with applications for a variety of quantum systems. Detailed benchmarks for different fidelity inhomogeneities, number of qubits for various rule and RL based algorithms underscore the results and show the superiority of the transformer-on-QuPairs method.

**Strengths:**

The paper outlines an important and extremely challenging problem which is physically motivated and underscored with a detailed physics model in the Appendix. The system level optimisation approach for optimising dynamic resource scheduling has not been tackled in the literature previously and this therefore makes this a valuable contribution. Moreover, the difficulty of the problem is well put by framing it as a more complicated version of the MWCSP. The applicability of this to various hardware architectures (photonics, ions, atoms, spins etc..) for networked quantum computing architectures also underscores the relevance of this work.

Moreover, the application of a transformer with different encodings (pre-information, dynamic and position) is a creative approach to solve a challenging problem in the control and design of large scale quantum experiments which has not emerged previously in the literature.

The benchmarks provided are comprehensive and clear. Table 1,2 an Figure 4 show very clear advantages of using the transformer approach which becomes increasingly more advantageous as the qubit number increases and the environment becomes more inhomogeneous (sigma(F) increases).

**Weaknesses:**

The scaleability and comparison in training times over different algorithms is largely omitted. A more detailed analysis and perhaps discussion of the limitations of particular approaches as the number of qubits scale would be extremely helpful, particularly for the Results shown in Table 3. This would make the analysis of the experiment more complete and strengthen the overall paper.

The manuscript is not as clear as it could be and the framing of the problem is not as strong as it could be either. In the abstract the authors claim a 3x improvement over rule based agents but no reference to a metric with which this improvement is quantified is made. Moreover, the authors use a quoted upper bound in fidelity from IBM ( a superconducting device), but the general networked architecture seems to be based on "a quantum control architecture tailored to a unique class of quantum resources featuring a spin-photon (Appendix A.8, Fig.6) interface conducive to remote entanglement routing" and has applications to trapped ions or neutral atoms. Moreover, the reference (IBM Website) does not provide any details on this upper bound used in the manuscript. Fidelities for networked architectures generally seem to be lower than this as shown in Ref. https://journals.aps.org/prl/abstract/10.1103/PhysRevLett.124.110501 (fidelity of 94%) and so some more justification or a clearer explanation would strengthen the manuscript. Additionally, the authors work simulates real physical interactions and the success of the proposed Transformer-on-QuPairs model relies on inhomogeneity in the fidelities (i.e. a high sigma(F)), but little discussion, of why such inhomogeneity arises in physical systems, nor any reference to existing work which shows qubit inhomogeneity is made. Providing a stronger case for the relevance of the highly inhomogeneous fidelity scenario would strengthen the benchmarks significantly.

The results in table 2 are somewhat confusing. It seems as though for a perfectly homogeneous fidelity, the different methods should perform exactly the same, but there are small but non-negligible differences. Is this a real result? Given the two uncertain (two sigma deviation) is larger than the absolute differences across methods it is not clear how to interpret these results. Some more clarity from the authors would help.

**Questions:**

What are the limitations of specific approaches when scaling the number of qubits in terms of runtime/memory, and how do they affect the results presented in Table 3?

How does the manuscript account for the inhomogeneity in fidelities that the Transformer-on-QuPairs model advantage relies on, and what causes this inhomogeneity in real quantum systems? Could examples and explicit references be provided.

Can the authors further justify the use of an upper bound in fidelity from superconducting qubits for a networked architecture?

Why do the methods in Table 2 show small differences in performance even for perfectly homogeneous fidelities, and how should these results be interpreted given the uncertainties?

Why is the static minimum spanning tree (MST) approach... anticipated to be an effective heuristic when the quantum system’s coherence time is indefinitely long or has a deterministic success probability during entanglement attempts.? Could the authors elaborate on this.

---

> ### Author Response · Authors · 2024-11-27
> **We thank the reviewer for the thoughtful comments. Please find our individual responses below.**
>
> Due to the characters limit, this response is divided into two comments. Please see subsequent comments for complete answers.
>
> ---
> **Q1: What are the limitations of specific approaches when scaling the number of qubits in terms of runtime/memory, and how do they affect the results presented in Table 3?**
>
> A1:
> We acknowledge that this work has limitations regarding the scalability of large sequence attention when increasing the number of qubits ( $N_q^2$). As discussed in Section 7 of the manuscript, scaling to a significantly larger Nq becomes computationally challenging due to the increased memory requirements of the transformer architecture. Specifically, this manuscript’s experiments were conducted on a single NVIDIA A30 GPU with 24GB of memory, which supports up to  $N_q$ = 160.
>
> However, the scalability of the transformer architecture benefits from advancements in hardware and large language model research. For example, recent progress has demonstrated the capability to handle inputs up to two million tokens [1] using GPU/TPU clusters, which corresponds to a theoretical  $N_q$  = 1414  for qubit pair attention. Such scalability is feasible with current hardware resources and is sufficient to cover most near-term NISQ (Noisy Intermediate-Scale Quantum) devices.
>
> For larger-scale systems beyond  $N_q$ = 1414, additional computational resources would be required to generate and train on networks of that size. Therefore, while the results in Table 3 reflect the computational limits of our current resources, the architecture remains flexible and adaptable to future scaling as more advanced hardware becomes available.
>
>
> Reference:
>
> [1]. https://deepmind.google/technologies/gemini/pro/
>
> ---
>
> **Q2: How does the manuscript account for the inhomogeneity in fidelities that the Transformer-on-QuPairs model advantage relies on, and what causes this inhomogeneity in real quantum systems? Could examples and explicit references be provided.**
>
> A2: Thank you for this insightful question and for highlighting this point. The inhomogeneity in fidelities arises from variations in the fabrication and control processes of real quantum systems. For example:
>
> 1.	Solid-state qubit systems such as superconducting qubits [1,2] and solid-state color center qubits [3] experience device-level variations during fabrication. These variations result in differences in the fidelity of single-qubit and two-qubit gates across the system.
>
> 2.	Trapped ion systems [4] and neutral atom systems [5] encounter variations in optical control across different spatial locations. This spatial variation introduces differences in operational fidelities.
>
> This inhomogeneity in fidelities is an intrinsic feature of real quantum systems. Our framework leverages this variation by optimizing the usage of components with better fidelity properties. By doing so, it enhances system performance without requiring any changes to the underlying hardware.
>
> References:
>
> [1]. https://quantum.ibm.com/services/resources
>
> [2]. Arute, Frank, et al. “Quantum supremacy using a programmable superconducting processor.” Nature 574.7779 (2019): 505-510.
>
> [3]. Li, Linsen, et al. “Heterogeneous integration of spin–photon interfaces with a CMOS platform.” Nature (2024): 1-7.
>
> [4]. https://ionq.com/quantum-systems/compare
>
> [5]. Bluvstein, Dolev, et al. “Logical quantum processor based on reconfigurable atom arrays.” Nature 626.7997 (2024): 58-65.
>
> ---
> **Q3: Can the authors further justify the use of an upper bound in fidelity from superconducting qubits for a networked architecture?**
>
> A3: Thank you for raising this important question. The maximum fidelity value (99.8%) used in the manuscript simulations is grounded in practical achievements across various state-of-the-art quantum platforms, including superconducting qubits [1], trapped-ion qubits [2], neutral-atom qubits [3], and diamond color centers [4]. This value represents a feasible upper bound for current quantum hardware technologies. Furthermore, real fidelity distributions for quantum systems can often be obtained from publicly available resources provided by quantum computer manufacturers [1].
>
> The simulations in the manuscript illustrate the applicability of our framework to realistic quantum systems. The assumed maximum fidelity serves as a practical benchmark for analyzing networked architectures. For specific implementations, the fidelity map and maximum values can be customized based on the properties of the target quantum hardware platform, ensuring that the framework remains relevant and effective for different systems.
>
> References:
>
> [1]. https://quantum.ibm.com/services/resources.
>
> [2]. https://ionq.com/quantum-systems/compare.
>
> [3]. Evered, Simon J., et al. “High-fidelity parallel entangling gates on a neutral-atom quantum computer.” Nature 622.7982 (2023): 268-272.
>
> [4]. Bartling, H. P., et al. “Universal high-fidelity quantum gates for spin-qubits in diamond.” arXiv preprint arXiv:2403.10633 (2024).

---

> > ### Author Response · Authors · 2024-11-27
> > **Individual responses (Continuous)**
> >
> > **Q4: Why do the methods in Table 2 show small differences in performance even for perfectly homogeneous fidelities, and how should these results be interpreted given the uncertainties?**
> >
> > A4: Thank you for raising this important point. The small differences in performance reported in Table 2 are primarily due to the stochastic nature of the Monte Carlo simulation used to model probabilistic events. In these simulations, we run 100 probabilistic trials and calculate the average performance across them. Consequently, even for the same method, the results are subject to inherent variability arising from the random sampling process. To account for this variability, we provide uncertainty intervals corresponding to two standard deviations (95% confidence level) to represent the expected range of error. These uncertainties are essential for interpreting the results and understanding the performance differences between methods.
> >
> > To better illustrate the comparative benefits of the Transformer-on-QuPair architecture versus the Greedy-on-QuPair architecture, we refer to Figure 4b in the manuscript. On average, the Transformer-on-QuPair architecture demonstrates a 3x improvement ($2^{\Delta \bar{\mu}}$) compared to the rule-based Greedy-on-QuPair method. However, due to the inherent uncertainties, individual sample values may vary, and the performance difference will follow a Gaussian distribution. The variance of this difference is influenced by the combined variances of the two methods, as shown in Table 2. Understanding these variations is key to interpreting the performance deltas in realistic scenarios.
> >
> > ---
> >
> > **Q5: Why is the static minimum spanning tree (MST) approach... anticipated to be an effective heuristic when the quantum system’s coherence time is indefinitely long or has a deterministic success probability during entanglement attempts.? Could the authors elaborate on this.**
> >
> > A5: When the quantum system’s coherence time is indefinitely long or the success probability during entanglement attempts is deterministic, the problem becomes equivalent to a static scheduling problem. In such scenarios, taking longer to establish entanglement does not impact fidelity because the system’s coherence time is effectively infinite. Alternatively, if entanglement can be established instantaneously (deterministic success probability p = 1) or the decoherence time is negligible compared to the entanglement building time, decoherence errors are effectively eliminated. In these cases, the only source of error is the intrinsic error of the two-qubit gate operations.
> >
> > The minimum spanning tree (MST) approach guarantees the minimum weight sum to connect all nodes in a graph. In the context of our qubit graph, this weighted sum corresponds to the cumulative error of the quantum system, assuming decoherence errors are negligible. For further optimization when only a subset of k qubits needs to be connected, the problem transforms into the k -MST problem, where the goal is to find the minimum-weight spanning tree for k vertices. Efficient approximation algorithms, such as greedy search, can be employed to solve the k-MST problem effectively in these cases.

---

> ### Comment · Reviewer_tMcL · 2024-12-02
>
> Thank you for providing these comments. I think the framing of the differences in performance in Table 2 could be portrayed clearer as explained in the comment.
>
> Following XEmk's review on the framing of the scheduling problem as NP-Hard, I have adapted my soundness and confidence rating. Complexity theory does not fall into my area of expertise, but I side somewhat with the reviewer and think the treatment of the complexity of the problem is not super clear, and the assignment a weight of zero to all terminal nodes is confusing.

---

### Official Review · Reviewer_XEmk · 2024-11-01

**Soundness:** 1
**Presentation:** 3
**Contribution:** 2
**Rating:** 3
**Confidence:** 3

**Summary:**

This work addresses the challenge of optimizing quantum resource scheduling in inhomogeneous systems. The authors formulate the scheduling problem as a dynamic minimum weight connected subgraph problem (MWCSP), a known NP-hard problem, and design a reinforcement learning (RL) framework, Transformer-on-QuPairs, to efficiently tackle this problem. Using a simulated environment, they benchmark the RL-based approach against rule-based algorithms, achieving a reported 3× improvement in performance.

**Strengths:**

This paper provides a robust solution to the quantum resource scheduling problem, readily applicable to experimental quantum computing frameworks, and easily generalizable to different hardware architectures. This makes the work highly relevant to the quantum computing community, and has potential for widespread application in enhancing the performance of future quantum systems.

**Weaknesses:**

While the authors claim to tackle an NP-hard problem, the lack of comparison with existing combinatorial optimization solvers is a notable limitation. A range of algorithms and solvers—such as traditional methods (e.g., simulated annealing, parallel tempering), commercial solvers (e.g., Gurobi, Hexaly), physics-inspired approaches (e.g., Ising machines, memcomputing), quantum solvers (e.g., D-Wave), and graph neural network-based machine learning solvers—are commonly used in related optimization contexts.  Yet, the paper benchmarks only against a few rule-based algorithms, which are generally not known to effectively solve NP-hard problems. The omission of comparisons to such established solvers limits the ability to assess the true novelty and strength of the proposed RL approach.

In addition, real-world experimental validation (beyond simulations) could further strengthen the results.

**Questions:**

1.	The authors suggest that the problem they address is at least as hard as MWCSP, an NP-hard problem. However, the cited MWCSP reference (Haouari et al., 2013) includes positive and negative weights, which prevent reduction to a minimum spanning tree problem. In contrast, the quantum resource scheduling problem here does not appear to involve negative weights. While the problem’s probabilistic nature may indeed make it more challenging—potentially classifying it as BPP, AM, or MA—could the authors provide a more rigorous explanation of why this specific problem is NP-hard?
2.	Can the authors benchmark their algorithm against a few established combinatorial optimization solvers?
3.	The authors assume an all-to-all qubit connectivity; however, many quantum computing architectures are locally connected. Can the authors comment on the generalizability of their approach to architectures with different connectivity structures?

---

> ### Author Response · Authors · 2024-11-27
> **We thank the reviewer for the thoughtful comments. Please find our individual responses below.**
>
> Due to the characters limit, this response is divided into two comments. Please see subsequent comments for complete answers.
>
> ---
>
> **Q1: The authors suggest that the problem they address is at least as hard as MWCSP, an NP-hard problem. However, the cited MWCSP reference (Haouari et al., 2013) includes positive and negative weights, which prevent reduction to a minimum spanning tree problem. In contrast, the quantum resource scheduling problem here does not appear to involve negative weights. While the problem’s probabilistic nature may indeed make it more challenging—potentially classifying it as BPP, AM, or MA—could the authors provide a more rigorous explanation of why this specific problem is NP-hard?**
>
> A1: Thanks for the reviewer's careful reading of this cited MWCSP reference (Haouari et al., 2013) including positive and negative weights. In fact, The NP-hardness of the Minimum Weight Connected Subgraph Problem (MWCSP) doesn’t require the possibility of negative weights edge in the graph. The reference here they are adding the extra negative weight in nodes instead of edges to finish the reduction of a known NP-hard problem Steiner tree problem (STP) to the MWCSP.
>
> The key for their proof reduction is to restrict the searched graph terminal node to the same as the STP. We can also directly set this requirement for a reduced version of MWCSP instead of raising a negative terminal weight for the node in the cited paper (Haouari et al., 2013). We can review their proof here, their MWCSG is our MWCSP in the manuscript:
>
> >"To the best of our knowledge, the complexity status of the MWCSG has never been established. In this section, we show that, despite its deceptive simplicity, the MWCSG cannot be solved efficiently unless P = NP."
>
> >**Lemma 1.** The MWCSG is NP-hard
>
> >**Proof:** The proof is based upon reduction from the Steiner tree problem (STP) in graphs which is known to be NP-hard [1]. This problem is defined as follows. Assume that we are given a connected, undirected graph $G=(V, E)$, with a nonnegative weight ce associated with each edge $e∈E$. The node set $V$ is partitioned into two subsets $S$ (set of Steiner nodes) and $T$ (set of terminal nodes). The STP is to find a shortest tree that spans all the nodes in $T$, and possibly some additional nodes from $S = V$ \ $T$. Given an STP instance, the reduction to an MWCSG instance that is defined on the same graph and with the same edge costs is achieved by further defining for each terminal node $j∈T$ a weight $γ_j = -M$ (where $M$ is a very large nonnegative integer), and for each Steiner node a zero weight. Let $G’  = (V’, E’)$ denote the optimal solution of the derived MWCSG instance. We can make the following observations:
>
> >(i) $G'$ is connected.
> >(ii) $G'$ is acyclic (because the edge costs are nonnegative).
> >(iii) $T ⊆V'$. This is an immediate consequence of the large negative weights of the terminal nodes.
>
> >Hence, we see from (i) and (ii) that $G‘$ is a tree, and we deduce from (iii) that it covers all terminal nodes. Thus, the optimal solution of MWCSG is a feasible STP solution. Furthermore, the cost of this solution is $c^*-M|T|$ where $c^*$ is the cost of the tree that covers the terminal nodes and (possibly) some Steiner nodes. Clearly, $c^*$ is the value of the shortest tree that covers all the terminals, hence it is an optimal STP solution. Thus, if the MWCSG problem is solvable in polynomial time so is the STP.”
>
> To avoid using the negative terminal node weight $-M$, we can also define the weight that
>
> “For each terminal node $j⊄T$ a weight $γ_j = M$ (where $M$ is a very large nonnegative integer)”
>
> We can achieve the same effect as requiring the MWCSG to include the STP terminal nodes set $T$ here. So the conclusion of MWCSP remains NP-hard for the positive weight graph is still valid.
>
> Reference:
>
> [1] Hwang FK, Richards DS, Winter P. The Steiner tree problem. Amsterdam: North-Holland; 1992.

---

> > ### Author Response · Authors · 2024-11-27
> > **Individual responses (Continuous)**
> >
> > **Q2: Can the authors benchmark their algorithm against a few established combinatorial optimization solvers?**
> >
> > A2: Thank you for raising this important point. We have benchmarked our algorithm against several established combinatorial optimization solvers and provided additional results here extending Table 1 in the manuscript. Within the rule-based algorithm group, we compared methods including simulated annealing and parallel tempering. These methods were evaluated starting from both random initial guesses and from greedy initial guesses.
> >
> > The results indicate that while simulated annealing and parallel tempering outperform random guessing by significantly increasing the average value of $\mu$, neither method surpasses the Greedy-on-QuPairs algorithm, even when initialized with greedy initial guesses. This highlights that the Greedy-on-QuPairs algorithm is particularly efficient for this problem within the rule-based algorithm category.
> >
> > In contrast, for the ML-based algorithms, we observed that a fully connected neural network applied to QuPairs outperforms all rule-based approaches, underscoring the effectiveness of ML-based methods for addressing the complexity of this scheduling problem. Furthermore, the transformer architecture demonstrates superior scalability and performance, particularly for quantum problems. This is consistent with the architecture’s proven effectiveness in decoding surface codes, as demonstrated in Google’s recent Alpha Quantum paper [1].
> >
> >
> > | **Types**       | **Strategy**                            | **$\bar{μ}$**       |
> > |------------------|-----------------------------------------|------------------|
> > | **Rule-based**   | Random                                  | 3.85 ± 0.23      |
> > |                  | Simulated annealing (From random guess) | 5.79 ± 0.49      |
> > |                  | Parallel tempering (From random guess)  | 6.52 ± 0.42      |
> > |                  | Static Minimum Spanning Tree            | 10.51 ± 0.55     |
> > |                  | Parallel tempering (From greedy guess)  | 13.31 ± 0.78     |
> > |                  | Simulated annealing (From greedy guess) | 13.80 ± 0.78     |
> > |                  | Greedy-on-QuPairs                       | 13.90 ± 0.62     |
> > | **RL-based**     | Transformer-on-Qubit                   | 3.91 ± 0.31      |
> > |                  | Fully-connected-on-QuPairs             | 14.70 ± 0.72     |
> > |                  | Transformer-on-QuPairs                 | 15.58 ± 0.84     |
> >
> > These results highlight the effectiveness of different algorithms across both rule-based and ML-based categories. Thank you for pointing out the need for these benchmarks, which further demonstrate the relative strengths of our approach.
> >
> > Reference:
> >
> > [1] Bausch, Johannes, et al. “Learning high-accuracy error decoding for quantum processors.” Nature (2024): 1-7.
> >
> > ---
> >
> > **Q3: The authors assume an all-to-all qubit connectivity; however, many quantum computing architectures are locally connected. Can the authors comment on the generalizability of their approach to architectures with different connectivity structures?**
> >
> > A3: Thank you for raising this important question. The proposed framework is specifically optimized for all-to-all qubit connectivity, which is a common feature of several quantum computing platforms, such as trapped ions [1], neutral atoms [2], and solid-state color centers [3]. These platforms inherently support all-to-all connectivity, making the framework directly applicable to their architectures.
> >
> > For locally connected architectures, such as those based on superconducting qubits [4,5], the framework can be adapted by incorporating connectivity restrictions into the entanglement scheduling process. By including these restrictions, the solution can accommodate the constraints imposed by local connectivity while still providing effective scheduling and optimization.
> >
> > However, to achieve the best performance under such constraints, the framework would require retraining to account for the specific connectivity structure. This retraining step ensures that the approach remains generalizable and can be tailored to different hardware architectures while maintaining high performance.
> > We will highlight these points in the manuscript to clarify the adaptability of the framework to various quantum computing architectures.
> >
> > References:
> >
> > [1]. https://ionq.com/quantum-systems/compare
> >
> > [2]. Bluvstein, Dolev, et al. “Logical quantum processor based on reconfigurable atom arrays.” Nature 626.7997 (2024): 58-65.
> >
> > [3]. Li, Linsen, et al. “Heterogeneous integration of spin–photon interfaces with a CMOS platform.” Nature (2024): 1-7.
> >
> > [4]. https://quantum.ibm.com/services/resources
> >
> > [5]. Arute, Frank, et al. “Quantum supremacy using a programmable superconducting processor.” Nature 574.7779 (2019): 505-510.

---

> > > ### Comment · Reviewer_XEmk · 2024-11-27
> > >
> > > I thank the authors for their detailed response and the inclusion of new benchmarks against simulated annealing and parallel tempering. However, I must respectfully point out that the authors’ response to Q1 contains a fundamental misunderstanding.
> > >
> > > Specifically, Haouari et al. (2013) assigned a large negative weight $-M$ to all terminal nodes $T$ and a weight of zero to all Steiner nodes $S$. This formulation ensures that all terminal nodes are included, while some Steiner nodes may also be included. In contrast, based on my understanding, the authors have assigned a weight of zero to all terminal nodes $T$ and a large positive weight $M$ to all Steiner nodes $S$. This approach ensures that all Steiner nodes are *excluded*, while some terminal nodes may not be included. Consequently, this does not correctly reduce the MWCSP to the Steiner tree problem.
> > >
> > > To clarify further, Haouari et al. (2013) explicitly highlight the importance of unrestricted node and edge weights in their formulation. To quote from the paper:
> > >
> > > “An important distinctive feature of the model is that both the node weights and the edge weights are unrestricted in sign.”
> > >
> > > “Indeed, while Bellman-Ford’s algorithm requires that the graph includes no cycles of negative weight, Dijkstra’s algorithm is even more stringent as it requires that the arc/edge weights are nonnegative.”
> > >
> > > “By contrast, a glaring fact is that the variant of the shortest path problem, where the graph includes negative cycles, received scant attention. This might be due to the fact that this latter problem is known to be NP-hard and therefore cannot be solved efficiently unless P = NP.”
> > >
> > > Additionally, it appears that the deterministic version of the quantum resource scheduling problem can be addressed using the minimum spanning tree algorithm, which has polynomial complexity. This observation casts doubt on whether the problem the authors aim to solve is genuinely NP-hard. It may also explain the effectiveness of the greedy algorithm, which typically struggles with NP-hard problems.

---

> > > > ### Author Response · Authors · 2024-11-28
> > > > **Replied to XEmk**
> > > >
> > > > We appreciate the reviewer’s thoughtful comments. We acknowledge that the modification of “For each terminal node  $j \notin T$ , assign a weight  $\gamma_j = M$  (where  $M$  is a very large nonnegative integer)” is not particularly effective for reducing the problem.
> > > >
> > > > However, there are alternative approaches to achieve the reduction:
> > > >
> > > > ---
> > > >
> > > > **1.	Applying a Hard Constraint Instead of Negative Node Weights**
> > > >
> > > > Instead of using negative weights in the cost function for certain nodes, we can apply a hard constraint during the problem definition stage. This modification ensures that all terminal nodes are included in the solution. The proof in Haouari et al. (2013) demonstrates that adding negative weights on nodes helps enforce these constraints, but this approach can be replaced with explicit hard constraints.
> > > >
> > > > In the context of our framework, this adjustment can be implemented by modifying the functions  $f_1$  and $f_2$  in Figure 2a of the manuscript to incorporate these constraints directly into the quantum scheduler. This ensures that the hard constraints are respected during optimization.
> > > >
> > > > Here are some detailed comparison with Steiner Tree Problem
> > > >
> > > > **Steiner Tree Problem (STP)**
> > > >
> > > > - **Input**:
> > > >   A connected, undirected graph $ G = (V, E) $ with positive edge weights $ c_e \geq 0 $ for all $ e \in E $, and a set of terminal nodes $ T \subseteq V $.
> > > >
> > > > - **Objective**:
> > > >   Find a minimum-weight tree $ S \subseteq G $ that spans all terminal nodes $ T $, possibly including additional Steiner nodes $ V_S = V \setminus T $.
> > > >
> > > >
> > > > **Construct an Instance of the MWCSG**
> > > >
> > > > 1. **Use the same graph** $ G = (V, E) $ with the same positive edge weights $ c_e \geq 0 $.
> > > >
> > > > 2. **Assign zero weights** to all nodes (both terminal and Steiner nodes). This ensures all node weights are non-negative.
> > > >
> > > > 3. **Add a constraint**: All terminal nodes $ T $ must be included in any feasible solution.
> > > >
> > > > 4. **Objective**: Find a connected subgraph $ G' = (V', E') $ that:
> > > >
> > > >    - Minimizes the total weight:  $\sum_{e \in E'} c_e$
> > > >    - Satisfies:  $ T \subseteq V' \subseteq V \quad \text{and} \quad E' \subseteq E$
> > > >
> > > >
> > > > **Establish Equivalence Between the STP and MWCSG**
> > > >
> > > > -  1. Any solution to this MWCSG instance corresponds to a solution to the STP, and vice versa.
> > > > -  2. Both problems require finding a minimum-weight connected subgraph that spans all terminal nodes.
> > > > -  3. Since node weights are zero, the total weight is determined solely by the edge weights.
> > > >
> > > > This completes the proof of equivalence between the Steiner Tree Problem (STP) and the Minimum-Weight Connected Subgraph Problem (MWCSG) under the given construction, which shows the problem we are solving in the manuscript is at least NP-hard.
> > > >
> > > > ---
> > > >
> > > > **2.	Allowing Negative Weights for Specific Qubit Nodes**
> > > >
> > > > Alternatively, we could redefine the problem to allow negative weights on qubit nodes. While the original problem assumes all qubit nodes have zero weights, introducing negative weights forces the inclusion of specific qubits when optimizing the quantum system’s benefit. Notably, edge weights in our quantum system remain nonnegative with physical error meaning. The approach outlined in Haouari et al. (2013) is compatible with a nonnegative edge weights graph but just requires node weights to be negative to enforce inclusion constraints if a hard-constraint approach is not applied. It doesn't require the graph to include the negative edge cycles, which is a different discussion from the mentioned Bellman-Ford algorithm and Dijkstra's algorithm.
> > > >
> > > > ---
> > > >
> > > > **Reduction to the k-MST Problem**
> > > >
> > > > For the deterministic version of the problem, it reduces to the k -MST problem rather than the standard MST problem, as selecting all qubit nodes is not optimal for quantum volume calculations. The k -MST problem is known to be NP-hard due to the exponential growth in combinatorial selection complexity [1]. In the context of large-scale qubit systems, a subset of qubits must be selected to build the cluster graph instead of using all the qubits to build the cluster, aligning with the k -MST formulation.
> > > >
> > > > As shown in Table 1, static MST algorithms do not perform well compared to other rule-based methods for this problem. While the greedy method is computationally efficient for solving k -MST problems as a heuristic algorithm, further improvements are observed when leveraging ML-based algorithms, demonstrating their effectiveness over rule-based solutions.
> > > >
> > > > References
> > > >
> > > > [1].	Ravi, Ramamurthy, et al. “Spanning trees—short or small.” SIAM Journal on Discrete Mathematics 9.2 (1996): 178-200.

---

### Official Review · Reviewer_5YAT · 2024-11-02

**Soundness:** 3
**Presentation:** 2
**Contribution:** 2
**Rating:** 3
**Confidence:** 3

**Summary:**

This paper presents a reinforcement learning framework, "Transformer-on-QuPairs," for dynamic inhomogeneous quantum resource scheduling, addressing the challenge of optimizing quantum resource state construction in the face of qubit inhomogeneity and probabilistic control. The approach uses a digitized environment with Monte Carlo Simulation to train reinforcement learning agents, focusing on self-attention mechanisms for qubit pairs. The study highlights the potential of this framework for co-designing physical and control systems in quantum computing.

**Strengths:**

- The research studies a central challenge in quantum systems, which is the real-time estimation and control of inherently inhomogeneous and probabilistic quantum resources, making it highly relevant to current technological advancements.
- The proposed method achieves an improvement in quantum system performance over rule-based agents, demonstrating a substantial enhancement in efficiency.

**Weaknesses:**

Major Concerns:

- The proposed optimization framework requires the availability of pre-characterized system information. It is crucial for the authors to explain in more detail the process of acquiring this information and the associated resource expenditures, particularly when the framework is to be applied to an unknown quantum system. This transparency is essential for assessing the practicality and feasibility of the framework in real-world scenarios.
- The framework appears to require real-time characterization for the maximum cluster size and the error associated with each established entanglement. I am curious about the cost and difficulty of obtaining this information in real quantum systems.
- The description of the framework is unclear to me. The authors should not assume that readers are familiar with their terminology. There are several points that need to be clarified further. For instance, what is the state matrix? What is the relationship between the state matrix and the pre-characterized system information? What does "the scheduling event is complete" mean? How do $f_1$ and $f_2$ function, and what are their inputs? How is the reward for the agent calculated?
- I believe that more details on the training process need to be provided in the manuscript.

Minor comments:

- There are some misleading sentences in the related works section. For instance, the authors stated, "The Transformer model, for example, has been effectively used in various applications such as ... and quantum state reconstruction (Carrasquilla et al., 2019)." However, I do not believe that Carrasquilla et al. (2019) utilized the Transformer model in their method.

**Questions:**

Please see "Weaknesses" above.

---

> ### Author Response · Authors · 2024-11-27
> **We thank the reviewer for the thoughtful comments. Please find our individual responses below.**
>
> Due to the characters limit, this response is divided into three comments. Please see subsequent comments for complete answers.
>
> ---
>
> **Q1: The proposed optimization framework requires the availability of pre-characterized system information. It is crucial for the authors to explain in more detail the process of acquiring this information and the associated resource expenditures, particularly when the framework is to be applied to an unknown quantum system. This transparency is essential for assessing the practicality and feasibility of the framework in real-world scenarios.**
>
> A1: Thank you for highlighting this important consideration. For physical quantum hardware systems, calibration is a standard process that provides users with the error distribution for each physical link between qubits. For instance, platforms like IBM Q and Google’s superconducting qubits [1,2] routinely offer detailed fidelity data for gates across all accessible qubits. Similarly, for systems where users do not directly access the hardware, such as IonQ or QuEra systems [3,4], the quantum computer providers supply pre-characterized measurement data for all qubit gates.
>
> Leveraging this hardware-specific qubit error distribution is crucial for optimizing the performance of the system. The proposed framework is designed to utilize this pre-characterized information to achieve improved results. For an unknown quantum system, acquiring the required parameters involves executing quantum circuits and measurement sequences on the hardware to determine error distributions. These processes are standard in real-world quantum computing workflows, ensuring that the necessary system information is available for optimization when the hardware is operable.
>
> References:
>
> [1]. https://quantum.ibm.com/services/resources
>
> [2]. Arute, Frank, et al. “Quantum supremacy using a programmable superconducting processor.” Nature 574.7779 (2019): 505-510.
>
> [3]. https://ionq.com/quantum-systems/compare
>
> [4]. Bluvstein, Dolev, et al. “Logical quantum processor based on reconfigurable atom arrays.” Nature 626.7997 (2024): 58-65.
>
>
> ---
>
> **Q2: The framework appears to require real-time characterization for the maximum cluster size and the error associated with each established entanglement. I am curious about the cost and difficulty of obtaining this information in real quantum systems.**
>
> A2: Thank you for this insightful question. In measurement-based quantum computing (MBQC) [1], it is standard practice to perform measurements on intermediate qubits to extract the required system information. Similarly, for quantum error correction [2], continuous measurements of data qubits are routinely performed to identify the system’s error state. These practices are well-established in real quantum systems and provide the necessary data for real-time characterization.
>
> For obtaining information about established entanglement, methods such as Bell-state measurements can be used in spin-photon systems [3]. These measurements allow the characterization of the spin state through spin-photon entanglement. When scaling to larger systems, technologies like MEMS-based optical cross-connect arrays [4] can facilitate routing for optical measurements with single-photon detector arrays, making it feasible to collect the required information even in large-scale quantum systems.
> In summary, the characterization process required by the framework is achievable with existing techniques in quantum computing systems, both for small-scale and scalable implementations.
>
> References:
>
> [1]. Walther, Philip, et al. “Experimental one-way quantum computing.” Nature 434.7030 (2005): 169-176.
>
> [2]. “Suppressing quantum errors by scaling a surface code logical qubit.” Nature 614, no. 7949 (2023): 676-681.
>
> [3]. Humphreys, Peter C., et al. “Deterministic delivery of remote entanglement on a quantum network.” Nature 558.7709 (2018): 268-273.
>
> [4]. Kim, J., et al. “1100 x 1100 port MEMS-based optical crossconnect with 4-dB maximum loss.” IEEE Photonics Technology Letters 15.11 (2003): 1537-1539.

---

> > ### Author Response · Authors · 2024-11-27
> > **Individual responses (Continuous)**
> >
> > **Q3: The description of the framework is unclear to me. The authors should not assume that readers are familiar with their terminology. There are several points that need to be clarified further. For instance, what is the state matrix? What is the relationship between the state matrix and the pre-characterized system information? What does "the scheduling event is complete" mean? How do f1 and f2 function, and what are their inputs? How is the reward for the agent calculated?**
> >
> > A3: Thank you for raising these important points. Below, we clarify the key concepts and their roles in the proposed framework:
> >
> > **State Matrix ($M_S$):**
> >
> > - The state matrix  $M_S$  is an  $N_q \times N_q$ matrix, where  $N_q$  is the number of qubits in the system. It dynamically stores entanglement information:
> >   - Initially, all elements are set to  0.
> >   - If qubits  i  and  j  are successfully entangled,  $M_S(i, j)$ = $M_S(j, i)$ = 1 .
> >   - If an entanglement trial between i  and j  is ongoing,  $M_S(i, j)$ = $M_S(j, i)$ = 0.5 .
> > The state matrix evolves over time and is not related to the pre-characterized system information.
> >
> >
> > **Pre-characterized System Information:**
> >
> > - This information is stored separately in the fidelity matrix  $M_F$  and the entanglement rate matrix  $M_R$ , which represent the successful entanglement probability and entanglement fidelity between qubits  $i$  and  $j$ , respectively.
> >
> >
> > **“Scheduling Event is Complete”:**
> >
> >  - A scheduling event is considered complete when all available qubits are either actively attempting entanglement or idle with no further scheduling opportunities. By analyzing the state matrix, we identify idle qubits and attempt to schedule additional entanglement trials. If no further scheduling is possible at a given time step, the event is marked as complete.
> >
> >
> > **Function $f_1$:**
> >
> >   - Input: The state matrix  $M_S$ .
> >   - Output: A determination of whether additional entanglement trials are possible.
> >
> >     - $f_1$ analyzes the dynamically connected cluster graph represented by  $M_S$, identifying which qubits are entangled or attempting entanglement. If further entanglement trials are feasible,  $f_1$  updates the state matrix and forwards it to the reinforcement learning (RL) agent for scheduling suggestions. If no further trials are possible,  $f_1$ concludes that the scheduling event is complete.
> >
> > **Reinforcement Learning (RL) Agent:**
> >
> > - The RL agent takes the updated state matrix from  $f_1$  and outputs an action matrix prioritizing potential entanglement actions. The system implements these actions, updates  $M_S$ , and continues to the next iteration.
> >
> > **Function $f_2$:**
> >   - Input: The state matrix  $M_S$.
> >   - Output: The size of the maximum cluster built in the system.
> >
> >     - If the cluster size exceeds a predefined threshold,  $f_2$ triggers the calculation of the system reward.
> >
> > **Reward Calculation:**
> > - The reward is calculated after the Monte Carlo simulation is complete. The reward is defined as  $\mu = \log_2(V_Q)$ , where  $V_Q$  is the quantum volume metric. This value reflects the system’s performance and is used as feedback for the RL agent. Figure 3c in the manuscript illustrates  $\mu$  (red curve), with the maximum value serving as the reward for the agent.
> >
> >
> >
> > We hope this clarifies the framework and its components. Thank you for pointing out areas that required further explanation.

---

> > > ### Author Response · Authors · 2024-11-27
> > > **Individual responses (Continuous)**
> > >
> > > **Comment 1: I believe that more details on the training process need to be provided in the manuscript.**
> > >
> > > Answer 1: Thank you for highlighting the need for additional details about the training process. We agree that more comprehensive explanations will improve the clarity of the manuscript. Below, we provide further details, which are included as an additional paragraph in the section 5.1 RL-based strategies section in the revised manuscript.
> > >
> > > The training process for the Transformer neural network begins with an initialization phase where the network is pre-trained to mimic the outputs of the Greedy-on-QuPair algorithm. This provides a baseline for the network’s parameters. To introduce variability and enhance generalization, random variations are added to the network parameters. The training then proceeds iteratively, with the network updating its parameters based on the rewards obtained from Monte Carlo simulations. The goal of each update is to guide the network toward actions that maximize the reward. This iterative process continues for 3000 epochs.
> > > To improve scalability and training efficiency, the Transformer-on-Qupairs architecture leverages transfer learning. Specifically, the model trained for  $N_q$ = 40  qubits is used as the initial model for training the  $N_q$ = 80  model. Similarly, the  $N_q$ = 80  trained model serves as the starting point for training the  $N_q$ = 120 model. This progressive training approach significantly reduces the computational overhead and speeds up convergence for larger systems.
> > >
> > > ---
> > >
> > >
> > > **Comment 2: There are some misleading sentences in the related works section. For instance, the authors stated, "The Transformer model, for example, has been effectively used in various applications such as ... and quantum state reconstruction (Carrasquilla et al., 2019)." However, I do not believe that Carrasquilla et al. (2019) utilized the Transformer model in their method.**
> > >
> > > Answer 2: Thank you for the reviewer’s careful reading and feedback. You are correct that Carrasquilla et al. (2019) did not use the Transformer model in their approach, but rather a more general machine learning method. The reference to quantum state reconstruction using Transformers was intended to cite a different work [1], which specifically applies Transformer-based techniques. We replace the citation with the correct reference in the revised manuscript to avoid this misunderstanding. And we add the latest citation of Google’s Alpha Quantum result using a transformer for quantum error-correction code decoding in the related work here [2].
> > >
> > > References:
> > >
> > > [1]. Ma, Hailan, et al. “Tomography of Quantum States from Structured Measurements via quantum-aware transformer.” arXiv preprint arXiv:2305.05433 (2023).
> > >
> > > [2]. Bausch, Johannes, et al. "Learning high-accuracy error decoding for quantum processors." Nature (2024): 1-7.

---

### Official Review · Reviewer_p48y · 2024-11-04

**Soundness:** 2
**Presentation:** 2
**Contribution:** 3
**Rating:** 6
**Confidence:** 4

**Summary:**

The paper tackles the issue of quantum resource scheduling, i.e., adjusting the parameters of a quantum system. The approach used is based on reinforcement learning (RL) and applies a transformer to efficiently model qubit connections. Results from a small case study are presented, where the new approach outperforms its competition.

**Strengths:**

The problem is interesting, important, and well motivated.

The approach seems fitting and is compared with a reasonable amount of baselines. It achieves superior performance.

The plots are well presented.

**Weaknesses:**

In certain parts, the paper is unnecessarily hard to understand. These issues include:
- The handling of the issue of complexity is off. Reasons for NP-hardness are given that are no real reasons and do not imply any NP-hardness proof. "Quantum" does not automatically make things harder.
- The beginning of the introduction is not well connected to the main content of the paper and has a lot of trailing references to the appendix.
- All matrices appear to be named "M" with some index. The variable name "N" is similarly overused etc. A better naming scheme would greatly aid understanding.

In a similar vein, I would greatly recommend introducing the problems more formally. The structure of the RL problem is just assumed and roughly derived from the underlying physics problem. Why is there no mapping to the standard MDP definition? This makes it hard to grasp the core of the problem.

A more standard formulation would also allow to apply a much greater range of standard algorithm running on the problem.

Most importantly, while outperformance is measured, there is not given a succinct reason. How does the approach perform on the "basis problem" MWCSP? Why is it not tested there?

While only very few formal errors persist, all citations are formatted incorrectly.

**Questions:**

None.

---

> ### Author Response · Authors · 2024-11-27
> **We thank the reviewer for the thoughtful comments. Please find our individual responses below.**
>
> Due to the characters limit, this response is divided into three comments. Please see subsequent comments for complete answers.
>
> ---
>
> **Q1: The handling of the issue of complexity is off. Reasons for NP-hardness are given that are no real reasons and do not imply any NP-hardness proof. "Quantum" does not automatically make things harder.**
>
> A1: Thanks for the reviewer mentioning the NP-hardness proof of our problem. We believe our problem is NP-hard by comparing it to another simpler NP-hard problem: the Minimum Weight Connected Subgraph Problem (MWCSP). The reason for our problem is harder than MWCSP is stated in the manuscript Section 3 in Complexity of the cluster building scheduling problem paragraph.
>
> We would like to further prove that MWCSP is NP-hard with the cited reference in the manuscript (Haouari et al., 2013). In fact, The NP-hardness of the MWCSP doesn’t require the possibility of negative weights edge in the graph. The reference here they are adding the extra negative weight in nodes instead of edges to finish the reduction of a known NP-hard problem Steiner tree problem (STP) to the MWCSP.
>
> The key for their proof reduction is to restrict the searched graph terminal node to the same as the STP. We can also directly set this requirement for a reduced version of MWCSP instead of raising a negative terminal weight for the node in the cited paper (Haouari et al., 2013). We can review their proof here, their MWCSG is our MWCSP in the manuscript:
>
> >“To the best of our knowledge, the complexity status of the MWCSG has never been established. In this section, we show that, despite its deceptive simplicity, the MWCSG cannot be solved efficiently unless P = NP. ”
>
> >**Lemma 1.** The MWCSG is NP-hard
>
> >**Proof:** The proof is based upon reduction from the Steiner tree problem (STP) in graphs which is known to be NP-hard [1]. This problem is defined as follows. Assume that we are given a connected, undirected graph $G=(V, E)$, with a nonnegative weight ce associated with each edge $e∈E$. The node set $V$ is partitioned into two subsets $S$ (set of Steiner nodes) and $T$ (set of terminal nodes). The STP is to find a shortest tree that spans all the nodes in $T$, and possibly some additional nodes from $S = V$ \ $T$. Given an STP instance, the reduction to an MWCSG instance that is defined on the same graph and with the same edge costs is achieved by further defining for each terminal node $j∈T$ a weight $γ_j = -M$ (where $M$ is a very large nonnegative integer), and for each Steiner node a zero weight. Let $G’  = (V’, E’)$ denote the optimal solution of the derived MWCSG instance. We can make the following observations:
>
> >(i) $G'$ is connected.
> >(ii) $G'$ is acyclic (because the edge costs are nonnegative).
> >(iii) $T ⊆V'$. This is an immediate consequence of the large negative weights of the terminal nodes.
>
> >Hence, we see from (i) and (ii) that $G‘$ is a tree, and we deduce from (iii) that it covers all terminal nodes. Thus, the optimal solution of MWCSG is a feasible STP solution. Furthermore, the cost of this solution is $c^*-M|T|$ where $c^*$ is the cost of the tree that covers the terminal nodes and (possibly) some Steiner nodes. Clearly, $c^*$ is the value of the shortest tree that covers all the terminals, hence it is an optimal STP solution. Thus, if the MWCSG problem is solvable in polynomial time so is the STP.”
>
> To avoid using the negative terminal node weight $-M$, we can also define the weight that
>
> “For each terminal node $j⊄T$ a weight $γ_j = M$ (where $M$ is a very large nonnegative integer)”
>
> We can achieve the same effect as requiring the MWCSG to include the STP terminal nodes set $T$ here. So the conclusion of MWCSP remains NP-hard for the positive weight graph is still valid.
>
> Reference:
>
> [1] Hwang FK, Richards DS, Winter P. The Steiner tree problem. Amsterdam: North-Holland; 1992.

---

> > ### Author Response · Authors · 2024-11-27
> > **Individual responses (Continuous)**
> >
> > **Q2: The beginning of the introduction is not well connected to the main content of the paper and has a lot of trailing references to the appendix.**
> >
> > A2: Thank you for raising this important point. The introduction was designed to highlight both the significance and the challenges of the field of quantum information science (QIS). We aimed to emphasize the importance of large-scale, high-fidelity quantum resource states for quantum applications and to provide recent progress as the context for the problem studied in this paper. We also underlined the inherent inhomogeneity in quantum hardware as a key challenge to optimizing quantum resources, given the intrinsic variations in physical quantum systems.
> >
> > The appendix is intended to provide additional background for general readers who may not be familiar with certain technical aspects of the topic. However, we agree that the connection between the introduction and the main content can be improved. Below is the revised version of the first paragraph of the introduction, which has also been updated in the manuscript to improve clarity and accessibility:
> >
> > **Introduction (Revised):**
> >
> > >“Quantum Information Science (QIS) is an emerging field poised to revolutionize computation, communication, precision measurement, and fundamental quantum science. At the heart of QIS lies the quantum resource state, which underpins quantum information representation and processing. For this paper, a quantum resource state refers to an entangled network of qubits (Appendix A.5, A.6). Achieving larger, high-fidelity quantum resource states is critical for advancing applications in material and drug discovery, optimization, and machine learning via quantum computing (Appendix A.6). Scaling physical qubit resources to meet the demands of quantum information processing is increasingly enabled by advances in solid-state quantum systems such as color centers and quantum dots (Appendix A.3). These systems leverage modern semiconductor fabrication technologies and heterogeneous integration (Wan et al., 2020; Li et al., 2024; Clark et al., 2024; Golter et al., 2023; Starling et al., 2023; Palm et al., 2023). Such technologies allow for large-scale quantum systems with dynamically configurable qubit interactions through remote entanglement (Humphreys et al., 2018), customized to meet system requirements (Choi et al., 2019; Nickerson et al., 2014; Nemoto et al., 2014). However, optimizing the control and scheduling of these large, complex systems is essential to maximize performance. Quantum resources exhibit inherent inhomogeneity due to their distinct physical properties and control mechanisms, which vary spatially and temporally. This inhomogeneity, coupled with the probabilistic nature of quantum operations like heralded remote entanglement (Appendix A.10), introduces stochastic challenges in error detection and system performance. These complexities render the optimization of quantum resource state construction an NP-hard problem. Nevertheless, achieving larger, high-fidelity quantum resource states offers exponential advantages in quantum information processing.”
> >
> > This revision strengthens the connection between the introduction and the core content of the paper, ensuring better clarity and flow for general and expert readers alike. Thank you for helping us improve this critical section.

---

> > > ### Author Response · Authors · 2024-11-27
> > > **Individual responses (Continuous)**
> > >
> > > **Q3: All matrices appear to be named "M" with some index. The variable name "N" is similarly overused etc. A better naming scheme would greatly aid understanding.**
> > >
> > > A3: Thanks for the reviewer pointing out this important point. We are using $M$ for the matrix and $N$ for numbers with different subscripts to classify the differences. We summarize all the $M$ and $N$ used in the manuscript here.
> > >
> > > $M$ variables:
> > >
> > > - $M_A$: Action matrix
> > >
> > > - $M_R$: Entanglement rate matrix
> > >
> > > - $M_F$: Fidelity matrix
> > >
> > > - $M_S$: State matrix
> > >
> > > $N$ variables:
> > >
> > > - $N_q$: Number of qubits in the system
> > > - $N_{max}$: Maximum cluster of the system
> > > - $N_t$: Number of the entanglement time step
> > > - $N_{dim}$: Dimension of the input for the token vector
> > >
> > > ---
> > >
> > > **Comment 1: I would greatly recommend introducing the problems more formally. The structure of the RL problem is just assumed and roughly derived from the underlying physics problem. Why is there no mapping to the standard MDP definition? This makes it hard to grasp the core of the problem.**
> > >
> > > A1: Thank you for this insightful question and for highlighting the need for a more formal mapping. Below, we provide the mapping of the problem to the standard Markov Decision Process (MDP) definition:
> > >
> > > **1. $S$ : State Space**
> > > - The state space is represented by the qubit graph, encoded as the state matrix  $M_S$ , which captures the current connectivity of qubits. Each element of  MS  reflects whether two qubits are connected, idle, or in the process of attempting entanglement.
> > >
> > > **2. $A$ : Action Space**
> > > - The action space corresponds to the entanglement actions between pairs of qubits that are idle (available for entanglement) but not yet connected in the qubit graph. Actions are represented by the action matrix  $M_A$, which specifies the pairs of qubits selected for entanglement attempts.
> > >
> > > **3. $R$ : Reward**
> > > - The reward for the system is the logarithm of quantum volume $\mu$, defined as  $\mu = log_2(V_Q)$  in the manuscript. This reward is calculated once the entanglement process is complete, as determined by the historical event of changing the state matrix $M_S$.
> > >
> > > **4. $P(S’ \| S, A)$ : Transition Dynamics**
> > > - The transition dynamics are governed by the success rate matrix $M_R$, which provides the success probabilities of entanglement attempts between specific pairs of qubits. The transitions are simulated using a Monte Carlo method, as described in the manuscript, to capture the probabilistic nature of the entanglement process.
> > >
> > > We hope this formal mapping clarifies the structure of the RL problem and highlights how it aligns with the standard MDP framework. Thank you for the opportunity to improve the clarity of our presentation.
> > >
> > > ---
> > >
> > > **Comment 2: Most importantly, while outperformance is measured, there is not given a succinct reason. How does the approach perform on the "basis problem" MWCSP? Why is it not tested there? While only very few formal errors persist, all citations are formatted incorrectly.**
> > >
> > > A2: Thank you for raising this point. The MWCSP is used primarily to evaluate problem complexity rather than the specific problem targets addressed by our framework. Our approach is designed to tackle a distinct class of quantum problems characterized by probabilistic success events during graph construction, which are inherently more complex than the MWCSP problem. The succinct reason for our framework’s outperformance lies in its ability to leverage the neural network’s capacity to evaluate long-term benefits through attention mechanisms and qubit pair prioritization, rather than relying on local optimization at each individual action step.
> > >
> > > We also appreciate the reviewer’s attention to the formatting of the citations. We already updated all the citations in the revised manuscript. Thank you again for pointing this out.

---

### Meta-Review · Area_Chair_wNxc · 2024-12-08

**Metareview:**

This paper addresses quantum resource scheduling using a reinforcement learning framework and a Transformer-on-QuPairs model to optimize qubit interactions. While the problem is timely and the approach demonstrates notable performance improvements over rule-based methods in simulations, the submission has significant shortcomings. The formulation lacks rigor, particularly in mapping the problem to a standard MDP framework. Additionally, comparisons with combinatorial optimization solvers are missing (see for instance https://arxiv.org/abs/2405.13947 and references therein), and real-world feasibility needs more justification. Thus, despite its potential, the work needs improvements for publication.

**Additional Comments On Reviewer Discussion:**

NA

---

### Decision · Program_Chairs · 2025-01-22

Reject